# Reflection Pretraining Enables Token-level Error Correction in Biological Sequence Generation

## Abstract

Large language models with natural language (e.g. English) have shown that generating **auxiliary tokens**—intermediate outputs not part of the final answer—enables powerful capabilities such as error correction, self-reflection, and more reliable reasoning. Methods like **Chain-of-Thought prompting** exploit the high expressiveness of natural languages (e.g., English), allowing models to verbalize internal states and perform complicated reasoning over text space. In contrast, **biological sequence models** (e.g., proteins, RNA, DNA) operate over token spaces with limited expressiveness , restricted to amino acids tokens or nucleotides tokens. As a result, these models lack mechanisms for externalized reasoning and are confined to producing **only** final sequence tokens without self-correction. In this work, we introduce **Bio-reflection pretraining**, a new framework that augments biological sequence models with an auxiliary `<reflect>` token. We select the reflection token because it provides flexible way for token-level modifications—such as error flagging, correction, swapping, and deletion—that directly target the types of mistakes most common and most urgent in biological sequence generation. By injecting synthetic errors during training and requiring the model to explicitly mark and correct them, we teach the model to engage in reflection and self-error-correction. This approach increases the effective expressiveness of biological sequence languages, enabling intermediate reasoning steps previously unattainable in this domain. We evaluate our method on the challenging task of *de novo peptide sequencing*, where intermediate reasoning is critical and the ground-truth label is unique and clearly defined. We demonstrate both theoretically and empirically that reflection pretraining substantially improves model accuracy on this task and enhances robustness against overfitting. Beyond accuracy gains, our framework enables **human-in-the-loop interaction**, allowing experts to guide or override reflection points during sequence generation. Taken together, reflection pretraining offers a principled path toward more interpretable and steerable biological sequence models, narrowing the gap between natural language models and their biological counterparts. **Codes, model outputs and model weights are available at** *Anonymous GitHub repository* for **full reproducibility**.

## 1 Introduction

Deep learning (LeCun et al., 2015) has significantly advanced the field of biology, with an increasing number of neural models being trained to generate and predict biological sequences such as DNA (Alipanahi et al., 2015; Zhang et al., 2021), RNA (Yang et al., 2022; Deng et al., 2022), and proteins (Xiao et al., 2025; Rives et al., 2021; Rao et al., 2021; Jumper et al., 2021; Lin et al., 2023). However, current biological sequence-generation models are constrained to produce only **answer tokens** directly related to specific tasks (e.g., drug design, de novo sequencing). This generation paradigm mirrors **conventional** natural language processing models (Naveed et al., 2023) before LLMs, where outputs are limited to final answers without intermediate reasoning or deliberation.[1] Recent work (Wei et al., 2022; Zhang et al., 2024a) has demonstrated that this answer-only generation

---

[1]For example, a machine translation model outputs just "goodbye" for the input "au revoir," or a mathematical language model returns just answer "121" for the input "$11 \times 11$," without revealing intermediate steps.

approach is suboptimal, both in terms of theoretical expressiveness and empirical performance. While a full theoretical analysis is provided in the Appendix, the intuition is straightforward: solving complex tasks often involves trial-and-error, exploration and even initial incorrect outputs before arriving at a final solution. Models constrained to generate only final answers are fundamentally incapable of performing this kind of structured, exploratory computation and thus fail to handle complex solution discovery effectively.

Chain-of-Thought (CoT) (Wei et al., 2022) prompting fundamentally changes how answers are generated in natural language models. Traditional neural models directly map an input sequence to a sequence of **answer tokens**, expressed as: $x_i : x_n \Rightarrow$ <answer$_1$> <answer$_2$> $\cdots$ <answer$_m$>.

In contrast, CoT introduces interleaved **non-answer tokens** that enable intermediate reasoning (Wei et al., 2022):

$$x_i : x_n \Rightarrow \text{<answer}_1\text{>} [\text{non}-\text{answer}_1][\text{non}-\text{answer}_2] \text{<answer}_2\text{>} \cdots \text{<answer}_m\text{>}.$$

Although these non-answer tokens, [non−answer], are discarded in the final output, they significantly enhance the model's capabilities by enabling it to *perform iterative computation, correct earlier errors, and reason across multiple steps* before reaching solutions.

This augmentation enables natural language (e.g. English) models to perform human-like reasoning such as self-correcting. Under suitable assumptions, CoT-enhanced models can even approach the theoretical upper bound of Turing completeness (Li et al., 2024) (detailed in Appendix), a level of expressiveness unattainable by direct-answer generation alone (Delétang et al., 2022).

However, such a reasoning process is only seen in LLMs due to the expressive capacity of human natural languages (e.g., English). Highly expressive languages can encode rich, structured information from hidden states **h** into tokens, including reflections on prior errors, intermediate computations, and procedural knowledge. In contrast, language like protein (Rives et al., 2021; Yilmaz et al., 2022) or RNA (Yang et al., 2022) relies on token and grammar systems with limited expressiveness, restricting models to **output only the next amino acid or nucleotides**. **In this paper, we formally demonstrate that the expressiveness of the foundational language directly affects the theoretical upper bound on the model's overall reasoning capability.**

In this work, we augment biological sequence models with *non-answer reflection tokens* through a novel **reflection-based pretraining** paradigm. We introduce a dedicated <reflect> token, chosen because it allows token-level modifications—such as error flagging, correction, swapping, and deletion—which directly address the types of mistakes most common and most critical in biological sequence generation. Unlike conventional pretraining, which limits models to only predicting sequence answer tokens, our approach explicitly equips models with the ability to generate reflection outputs that capture error-corrective signals. We focus on the challenging task of *de novo* protein sequencing, where the space of possible outputs is vast, yet the ground-truth label is *unique* and *well defined*—making it an ideal benchmark for testing reasoning-based biological approaches. We construct a large-scale and carefully curated reflection dataset, specifically designed to simulate error-prone predictions. Within this framework, we introduce two complementary mechanisms for error injection, enabling the model to encounter realistic mistakes and learn to *self-reflect* on its predictions. By training the model to detect, annotate, and revise these errors through reflection tokens, we show that biological sequence models can acquire a structured process of *self-correction*, thereby improving both accuracy and interpretability compared to standard pretraining.

**Our extensive experiments yield the following key findings**: **(1)** With appropriate training techniques and data augmentation, protein sequence models can perform **self-correction and self-reflection** capabilities that go beyond simply generating just answer tokens; **(2)** Finetuning alone does not confer reflection ability in bio-models; such capacity **emerges only through pretraining**; **(3)** Within a certain threshold, increasing the number of manually injected errors during training improves the model's ability to **self-correct**; **(4)** Reflection pretraining leads to significant accuracy gains, driven by both **enhanced generational ability** and a **counter-overfitting** effect during training and error-data augmentation; **(5)** The resulting models enable **human-in-the-loop generation**, allowing experts to actively engage with reflection and error-correction steps. This interactivity brings biological language models closer in capability and usability to large natural language models.

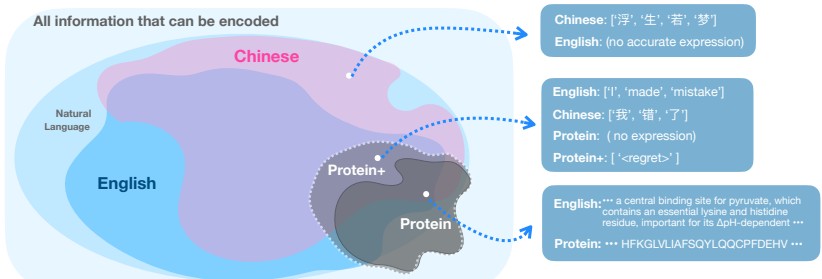

Figure 1: Expressiveness of Languages.

## 2 BIO LANGUAGE EXPRESSIVENESS

### 2.1 REFLECTION, ERROR-CORRECTION, AND THE LIMITS OF BIOLOGICAL TOKENS

Neural network expressiveness depends strongly on the tokens a model can output. Biological sequence models generate only **answer tokens** (amino acids or nucleotides), forcing them to commit to final sequences without intermediate reasoning. This prevents (1) multi-step deliberation, (2) flexible computation, and (3) use of externalized memory.

Natural language models overcome these limitations using **auxiliary reasoning tokens** (e.g., Chain-of-Thought), which allow intermediate thoughts, checks, and revisions. Natural languages are expressive enough to encode logic, heuristics, and self-evaluation.

Biological languages lack such expressive capacity: amino-acid alphabets were shaped for biochemical function, not reasoning. They cannot express concepts like "this residue is incorrect" or "swap these two positions."

To address this, we introduce the notion of **biological language expressiveness** and augment biological models with an auxiliary token, `<reflect>`. Unlike amino-acid tokens, `<reflect>` encodes higher-level information—flagging likely mistakes and suggesting corrections, swaps, or deletions. These local edits are crucial because small token-level errors can significantly disrupt biological function. The reflection token provides a controlled mechanism for identifying and correcting such errors.

### 2.2 LANGUAGE EXPRESSIVENESS

We formalize a language as $\mathbf{L} = (\mathcal{G}, \mathcal{V})$, where $\mathcal{G}$ is its grammar and $\mathcal{V}$ its vocabulary. The set of meanings representable in the language is $\mathbb{S}_{\mathbf{L}}$. The **expressiveness** of $\mathbf{L}$ is:

$$\text{Expressiveness}(\mathbf{L}) := |\mathbb{S}_{\mathbf{L}}|.$$

Even with infinitely many valid sequences, languages differ in the meanings they can encode. If $\mathbb{S}_{\mathbf{L}_1} \subset \mathbb{S}_{\mathbf{L}_2}$, then $\mathbf{L}_2$ is strictly more expressive.

**Natural languages** (e.g., English, Chinese) are highly expressive due to their large vocabularies and flexible grammar, enabling complex reasoning, explanation, and abstraction.

**Biological languages** such as protein sequences, $\mathbf{L}_{\text{protein}} = (\mathcal{G}_{\text{protein}}, \mathcal{V}_{\text{protein}})$ with 20 amino acids, are excellent for encoding structure and function but extremely limited for reasoning. Statements like "I am wrong" have no analogue in $\mathbb{S}_{\text{protein}}$.

### 2.3 LANGUAGE EXPRESSIVENESS DETERMINES CoT EXPRESSIVENESS

Biological sequence models generate only answer tokens, so any internal signals—uncertainty, regret, or error detection—remain hidden and never appear in the output (Fig. 6). Thus, these models cannot externalize reasoning or perform self-correction.

Figure 2: Comparative framework for prompting in natural and biological language models.

In natural language models, Chain-of-Thought introduces **non-answer tokens** that serve as intermediate reasoning steps (Fig. 6). These allow the model to revise or expand its reasoning before committing to an answer.

> **Essence of CoT**
>
> CoT maps hidden states $\mathbf{h}$ into language tokens, $\mathbb{S}_\Theta \to \mathbb{S}_\mathbf{L}$, enabling signals such as "I made a mistake" to appear as non-answer tokens before final answers.

Because amino-acid or nucleotide alphabets cannot encode such concepts, the CoT expressiveness of biological models is effectively **zero**. The `<reflect>` token provides the necessary channel to express intermediate reasoning and perform explicit error correction.

# 3 REFLECTION PRETRAINING FOR PROTEIN (PEPTIDE) SEQUENCE PREDICTIONS

## 3.1 TASK CHOICE AND PROBLEM FORMULATION

To this end, we propose *reflection-pretraining* for protein sequencing models to allow CoT reasoning beyond generating solely answer tokens (amino acids). We choose the the task of **de novo peptide sequencing** (Fig. 3) for five key reasons: 1) It is a foundational problem in protein sequence prediction, as peptide sequencing remains the primary method for determining amino acid sequences from nature. 2) The task is reasoning-intensive, requiring the model to perform complex computations on spectrum signals-well aligned with the reflection paradigm. 3) Large-scale training data is readily available. 4) Evaluation is precise, with each input spectrum paired with a gold-standard sequence, unlike protein language modeling with ambiguous metrics. 5) Each token corresponds to a discrete reasoning step, enabling fine-grained, token-level reflection.

In the *de novo* peptide sequencing task (Fig. 3), the model is given a spectrum instance $\mathbf{H} = \{\mathbf{I}, c, m\}$, produced by a mass spectrometer (Fig. 3 when provided with a biological sample (Fig. 3. The spectrum consists of:

- $\mathbf{I} = \{(\text{m/z}_1, i_1), (\text{m/z}_2, i_2), \ldots, (\text{m/z}_k, i_k)\}$, a set of $k$ observed mass-to-charge and intensity pairs, filtered by a signal threshold, (Fig. 3 MS/MS)

- $c \in \mathbb{Z}^+$, the charge state of the precursor ion, and

- $m \in \mathbb{R}^+$, the total measured mass of the peptide.

The objective is to **predict the underlying amino acid (protein) sequence** $\mathbf{A} = \{a_1, a_2, \ldots, a_n\}$, where each token $a_i \in \mathcal{V}_{\text{protein}} = \{20 \text{ amino acids tokens}\}$ belongs to the vocabulary of the protein language $\boldsymbol{L}_{\text{protein}} = (\mathcal{G}_{\text{protein}}, \mathcal{V}_{\text{protein}})$. The mapping $\mathbf{H} \mapsto \mathbf{A}$ requires the model to reason over the structure and intensity patterns in $\mathbf{I}$ while conforming to biochemical constraints imposed by $c$ and $m$.

*Note that, although we describe our method in the context of the protein sequence space, it applies to other biological sequence prediction tasks-including RNA, DNA, and synthetic polymers.*

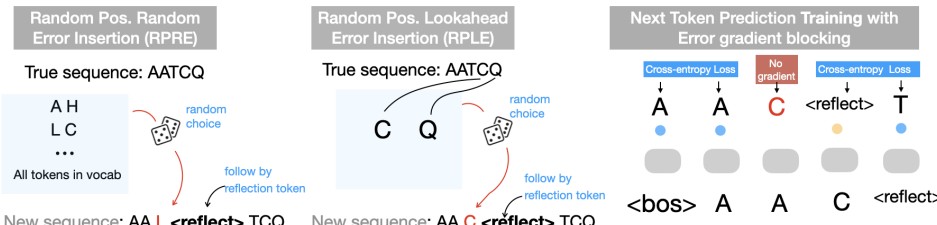

Figure 3: De novo peptide sequencing workflow using tandem mass spectrometry.

## 3.2 REFLECTION TOKEN AUGMENTED PROTEIN SEQUENCE FOR NEXT TOKEN PREDICTION TRAINING

One of the most powerful forms of reasoning is self-reflection and error correction. Prior work shows that sequence models often encode a sense of "regret" in their latent state $\mathbf{h}$-that is, the realization of an error *after* generating an answer token $\langle a \rangle$. In large language models (LLMs), the high expressiveness of the English language $\boldsymbol{L}_{\text{english}}$ enables this information to be verbalized, making reflection a highly effective mechanism for improving reasoning accuracy. However, in protein models, this is not possible: the latent regret state $\mathbf{h}_{\text{regret}}$ cannot be decoded into any token $a \in \mathcal{V}_{\text{protein}}$, i.e., $\mathbf{h}_{\text{regret}} \Rightarrow a$ is infeasible.

To enable self-reflection in protein sequence models, we augment the vocabulary with a reflection token, defining $\mathcal{V}_{\text{protein+}} = \mathcal{V}_{\text{protein}} \oplus \langle \texttt{reflect} \rangle$. We then construct a modified training dataset $\mathcal{D}_{\text{de novo+}}$ (Figure 3 ) by injecting synthetic errors (e.g., incorrect amino acids) into sequences and appending corrections using $\langle \texttt{reflect} \rangle$.

In this section, we introduce two novel strategies for error injection (Fig. 4), as well as our dynamic data updating and gradient blocking strategy for training protein models with reflection-based next-token prediction on the *de novo* sequencing task.

Figure 4: Error injection and reflection training for augmenting reasoning in bio. sequence models.

### 3.2.1 MODEL ARCHITECTURE

We adopt a Transformer-based encoder-decoder architecture, following standard designs for this task (Yilmaz et al., 2023). The input spectrum is denoted as $\boldsymbol{H} = \{\mathbf{I}, c, m\}$, where $\mathbf{I} = \{(m/z_k, i_k)\}_{k=1}^{K}$ consists of $K$ mass-to-charge and intensity pairs. Each $(m/z_k, i_k)$ is embedded via a sinusoidal encoding and projected into a $d$-dimensional vector. The resulting sequence of embeddings forms the encoder output $\boldsymbol{E} \in \mathbb{R}^{K \times d}$.

The decoder is an autoregressive Transformer with causal self-attention and cross-attention to $\boldsymbol{E}$. Let $\boldsymbol{h}_t^{(i)}$ denote the hidden state at decoding step $t$ and decoder layer $i$. Causal self-attention ensures that each position only attends to past outputs: $\boldsymbol{h}_t^{(i)} = \text{Attn}\left(\boldsymbol{h}_t^{(i-1)}, \{\boldsymbol{h}_{1:t-1}^{(i-1)}\}\right)$. Each $\boldsymbol{h}_t^{(i)}$ then attends to the spectrum features via cross-attention over $\boldsymbol{E}$. The final decoder output at step $t$, denoted $\boldsymbol{h}_t^{(L)}$, is projected onto the vocabulary $\mathcal{V}_{\text{protein+}}$: $P_t(\cdot \mid \boldsymbol{S}, \boldsymbol{y}_{<t}) = \text{softmax}\left(\boldsymbol{W} \boldsymbol{h}_t^{(L)}\right)$; Token $\boldsymbol{y}_t$ is sampled from $P_t(\cdot \mid \boldsymbol{S}, \boldsymbol{y}_{<t})$. Training proceeds via standard next-token prediction (NTP), minimizing the cross-entropy loss over ground-truth sequences.

### 3.2.2 RANDOM POSITION, RANDOM ERROR (RPRE) INJECTION

For a given input-spectrum and ground-truth sequence pair $(\boldsymbol{H}, \boldsymbol{A} = \{a_1, a_2, \ldots, a_n\}) \in \mathcal{D}_{\text{de novo}}$, we perform *reflection error injection* (RPRE) to construct training sequences for teaching the model how to reflect on its past generation and correct potential mistakes.

In the RPRE strategy, we randomly sample a position $t \sim \text{Uniform}(1, n)$ and a random token $\tilde{a}_t \sim \mathcal{V}_{\text{protein}}$. We then replace the token at position $t$ with $\tilde{a}_t$, regardless of correctness, and insert a reflection token $\langle\texttt{reflect}\rangle$ immediately afterward, followed by the original correct token $a_t$. The modified training target becomes:

$$\boldsymbol{A}' = \{a_1, \ldots, a_{t-1}, \tilde{a}_t, \langle\texttt{reflect}\rangle, a_t, a_{t+1}, \ldots, a_n\}$$

This approach enables the model to learn how to self-correct through reflection. Notably, when $\tilde{a}_t = a_t$ (i.e., the random token happens to be correct), the reflection becomes a no-op-teaching the model to maintain confidence when no error is present. As a result, RPRE encourages both correction and retention behavior through the same mechanism.

### 3.2.3 RANDOM POSITION, LOOKAHEAD ERROR (RPLE) INSERTION

While RPRE introduces reflection opportunities, the injected error $\tilde{a}_t$ may be overly implausible (e.g., "$1 + 1 = 3\langle\text{reflect}\rangle2$") and thus trivially detectable. To inject more cognitively challenging errors, we propose *Random Position, Lookahead Error (RPLE)* insertion, which samples replacement tokens within the ground-truth sequence itself.

Given a spectrum-label pair $(\boldsymbol{H}, \boldsymbol{A} = \{a_1, a_2, \ldots, a_n\}) \in \mathcal{D}_{\text{de novo}}$, we randomly select a position $t \sim \text{Uniform}(1, n)$ and choose an error token $\tilde{a}_t$ from a set of true tokens that appear later in the sequence: $\tilde{a}_t \sim \{a_{t+1}, a_{t+2}, \ldots, a_n\}$. We then replace $a_t$ with $\tilde{a}_t$, append the reflection token $\langle\texttt{reflect}\rangle$, and restore the correct token $a_t$. This strategy creates more realistic, sequence-consistent errors and helps the model learn to correct subtle confusions such as token swaps or premature token use–common in peptide prediction (e.g., predicting "BA" instead of "AB").

### 3.2.4 BATCH-LEVEL ONLINE DYNAMIC ERROR INJECTION

To prevent the model from memorizing the same sequence and error type and encourage generalizable reflection behavior, we adopt *batch-level online dynamic error injection* during training. For each mini-batch randomly sampled from $\mathcal{D}_{\text{de novo}}$, we apply error injection in real-time to the target sequences.

Given an injection ratio $\alpha \in [0, 1]$, a fraction $\alpha$ of sequences in each batch are modified using either the RPRE or RPLE strategy. The remaining $(1 - \alpha)$ fraction are left unchanged:

---

**Algorithm 1** Online Dynamic Reflection-Error Injection During Training

---

**Require:** Batch $\mathcal{B} = \{(\boldsymbol{H}^{(i)}, \boldsymbol{A}^{(i)})\}_{i=1}^B$, injection ratio $\alpha \in [0, 1]$
1: **for** each $(\boldsymbol{H}, \boldsymbol{A})$ in $\mathcal{B}$ **do**
2:     **if** $\text{Uniform}(0, 1) < \alpha$ **then**
3:         Sample $t \sim \text{Uniform}(1, |\boldsymbol{A}|)$, choose RPRE or RPLE
4:         $\tilde{a}_t \sim$ **RPRE**: $\mathcal{V}_{\text{protein}}$     **RPLE**: $\{a_{t+1}, \ldots, a_n\}$
5:         $\boldsymbol{A}' \leftarrow \{a_1, \ldots, a_{t-1}, \tilde{a}_t, \langle\texttt{reflect}\rangle, a_t, \ldots, a_n\}$
6:         Replace $\boldsymbol{A}$ with $\boldsymbol{A}'$ in $\mathcal{B}$
7:     **end if**
8: **end for**

---

### 3.2.5 ERROR POSITION GRADIENT BLOCKING

To ensure that injected errors act as *contextual signals* rather than learning targets, we apply **gradient blocking** at error positions during training. Formally, given a modified target sequence $\boldsymbol{A}' = \{a_1, \ldots, \tilde{a}_t, \langle\texttt{reflect}\rangle, a_t, \ldots\}$, we exclude the loss term at position $t$–corresponding to the injected error token $\tilde{a}_t$–from the training objective. This allows the model to condition on the error via causal attention and learn to generate the reflection token $\langle\texttt{reflect}\rangle$ in response, without learning the generation of the incorrect prediction itself (Fig. 4 right panel).

Table 1: Comparison of different settings of reflection pretraining and baseline on 9-Species test set.

| Method (beam size = 1) | Mouse | Human | Yeast | M.mazei | Honeybee | Tomato | R.bean | Bacillus | C.bacteria |
|---|---|---|---|---|---|---|---|---|---|
| **Standard Pretrain** | | | | Amino Acid Precision | | | | | |
| Transformer ($\Delta$ baseline) | 0.717 | 0.649 | 0.752 | 0.713 | 0.706 | 0.763 | 0.714 | 0.753 | 0.66 |
| **placeholder Token Baselines** | | | | | | | | | |
| Transformer + 60% Insertion | 0.704 | 0.633 | 0.755 | 0.692 | 0.680 | 0.745 | 0.701 | 0.726 | 0.630 |
| **Reflect. Finetune w/ *RPRE*** | | | | | | | | | |
| Transformer + 60% Error | 0.737 | 0.672 | 0.736 | 0.739 | 0.690 | 0.784 | 0.749 | 0.774 | 0.684 |
| **Reflect. Pretrain w/ *RPRE*** | | | | | | | | | |
| Transformer + 60% Error | 0.765 | 0.713 | 0.796 | 0.772 | 0.721 | 0.806 | 0.795 | 0.811 | 0.713 |
| **Reflect. Pretrain w/ *RP(RE +LE)*** | | | | | | | | | |
| Transformer + 60% Error | 0.784 | 0.735 | 0.808 | 0.786 | 0.743 | 0.813 | 0.808 | 0.820 | 0.723 |
| + 90% Error | **0.792** | **0.752** | **0.809** | **0.790** | **0.744** | **0.822** | **0.817** | **0.826** | **0.737** |
| | (+10.5%) | (+15.9%) | (+7.6%) | (+10.8%) | (+5.4%) | (+7.7%) | (+14.4%) | (+9.7%) | (+11.2%) |
| + 99% Error | 0.786 | 0.739 | 0.807 | 0.789 | 0.742 | 0.819 | 0.814 | 0.823 | 0.732 |
| **Standard Pretrain** | | | | Peptide Precision | | | | | |
| Transformer ($\Delta$ baseline) | 0.443 | 0.433 | 0.584 | 0.522 | 0.460 | 0.606 | 0.652 | 0.580 | 0.413 |
| **Reflect. Finetune w/ *RPRE*** | | | | | | | | | |
| Transformer + 60% Error | 0.443 | 0.434 | 0.589 | 0.530 | 0.465 | 0.607 | 0.563 | 0.583 | 0.416 |
| **Reflect. Pretrain w/ *RPRE*** | | | | | | | | | |
| Ours (60% ER, 6 AA/PEP) | 0.485 | 0.494 | 0.632 | 0.575 | 0.507 | 0.644 | 0.629 | 0.635 | 0.458 |
| **Reflect. Pretrain w/ *RP(RE +LE)*** | | | | | | | | | |
| Transformer + 60% Error | 0.513 | 0.535 | 0.651 | 0.592 | 0.534 | 0.658 | 0.665 | 0.662 | 0.470 |
| + 90% Error | **0.533** | **0.563** | **0.661** | **0.605** | **0.544** | **0.668** | **0.657** | **0.674** | **0.490** |
| | (+20.3%) | (+30.0%) | (+13.2%) | (+15.9%) | (+18.3%) | (+10.2%) | (+0.8%) | (+16.2%) | (+0.9%) |
| + 90% Error | 0.515 | 0.537 | 0.648 | 0.595 | 0.534 | 0.660 | 0.657 | 0.660 | 0.474 |

# 4 EXPERIMENTS

## 4.1 EXPERIMENTAL SETUP

**Datasets.** Following established protocols (Yilmaz et al., 2023), we use the MassIVE-KB dataset (Wang et al., 2018), which contains **30 million peptide-spectrum matches (PSMs)** collected from diverse mass spectrometry platforms. Evaluation is conducted on the widely adopted 9-species-v1 and 9-species-v2 benchmarks.

**Implementation.** Input peaks and amino acids are embedded into 512-dimensional vectors. Both the spectrum encoder and peptide decoder are 9-layer Transformers with 8 attention heads and 1024-dimensional hidden states. Models are trained on eight NVIDIA A100 80GB GPUs using AdamW ($\alpha_0 = 5 \times 10^{-4}$) with a 10k-step linear warmup followed by cosine decay.

**Evaluation Metrics.** We report two metrics. **Amino acid-level (AA) Precision** considers a prediction correct if the amino acid token aligns with the target position. Accuracy is defined as $M_{\mathrm{AA}}/T_{\mathrm{AA}}$, where $M_{\mathrm{AA}}$ is the number of matched amino acids and $T_{\mathrm{AA}}$ is the total number of predicted amino acids. **Peptide-level precision** considers a prediction correct only if the entire predicted peptide matches the ground truth sequence exactly. Precision is defined as $M_{\mathrm{pep}}/T_{\mathrm{pep}}$, where $M_{\mathrm{pep}}$ is the number of matched peptides and $T_{\mathrm{pep}}$ is the total number of predicted peptides.

## 4.2 RESULTS

**Results Analysis.** As shown in Table 1, reflection-based pretraining leads to substantial improvements in both amino acid and peptide precision across all 9 species. In contrast, **finetuning** shows negligible gains over baselines, and reflection tokens are never used during inference (Table 3), indicating that reflection behavior is not learned through finetuning alone.

Notably, increasing the error ratio from 60% to 90% further boosts peptide precision in all species, while also increasing the frequency of reflection token usage from 2.3% to 4.46% during inference (Table 3). Despite most `<reflect>` tokens retaining the original answer (e.g., 67.8% in the 90% error model), this indicates a learned capacity for self-assessment. Moreover, incorporating RPLE during pretraining introduces more realistic, confusing errors, pushing peptide precision further up.

Table 2: Comparison among bio-inspired methods and reflection reflection-pretrained model.

| Method | Mouse | Human | Yeast | M.mazei | Honeybee | Tomato | R.bean | Bacillus | C.bacteria | Average |
|---|---|---|---|---|---|---|---|---|---|---|
| **AA Precision** | | | | | | | | | | |
| **Bio. Inspired Methods** | | | | | | | | | | |
| Peaks | 0.600 | 0.639 | 0.748 | 0.673 | 0.633 | 0.728 | 0.644 | 0.719 | 0.586 | 0.663 |
| DeepNovo | 0.623 | 0.610 | 0.750 | 0.694 | 0.630 | 0.731 | 0.679 | 0.742 | 0.602 | 0.673 |
| PointNovo | 0.626 | 0.606 | 0.779 | 0.712 | 0.644 | 0.733 | 0.730 | 0.768 | 0.589 | 0.688 |
| InstaNovo | 0.703 | 0.636 | 0.691 | 0.712 | 0.660 | 0.732 | 0.711 | 0.739 | 0.619 | 0.689 |
| Casanovo, Beam = 1 | 0.717 | 0.649 | 0.752 | 0.713 | 0.706 | 0.763 | 0.714 | 0.753 | 0.663 | 0.704 |
| HelixNovo, Beam = 1 | 0.750 | 0.648 | 0.758 | 0.766 | 0.699 | 0.757 | 0.771 | 0.799 | 0.661 | 0.734 |
| **Reflection Pretrain** | | | | | | | | | | |
| 90% Error, Beam = 1 | **0.792** | **0.752** | **0.809** | **0.790** | **0.744** | **0.822** | **0.817** | **0.826** | **0.737** | **0.788** |
| 90% Error, Beam = 3 | **0.799** | **0.763** | **0.818** | **0.800** | **0.755** | **0.827** | **0.832** | **0.836** | **0.745** | **0.797** |
| 90% Error, Beam = 5 | **0.805** | **0.774** | **0.826** | **0.810** | **0.766** | **0.831** | **0.846** | **0.846** | **0.753** | **0.806** |
| **Peptide Recall** | | | | | | | | | | |
| **Bio. Inspired Methods** | | | | | | | | | | |
| Peaks | 0.197 | 0.277 | 0.428 | 0.356 | 0.287 | 0.403 | 0.362 | 0.387 | 0.203 | 0.333 |
| DeepNovo | 0.286 | 0.293 | 0.462 | 0.422 | 0.330 | 0.454 | 0.436 | 0.449 | 0.253 | 0.387 |
| PointNovo | 0.355 | 0.351 | 0.534 | 0.478 | 0.396 | 0.513 | 0.511 | 0.518 | 0.298 | 0.439 |
| InstaNovo | 0.471 | 0.455 | 0.559 | 0.528 | 0.466 | 0.732 | 0.564 | 0.576 | 0.416 | 0.530 |
| Casanovo | 0.443 | 0.433 | 0.584 | 0.522 | 0.460 | 0.606 | 0.652 | 0.580 | 0.413 | 0.521 |
| **Reflection Pretrain** | | | | | | | | | | |
| 90% Error, Beam=1 | **0.533** | **0.563** | **0.661** | **0.605** | **0.544** | **0.668** | **0.657** | **0.674** | **0.490** | **0.600** |
| 90% Error, Beam=3 | **0.540** | **0.573** | **0.669** | **0.615** | **0.555** | **0.674** | **0.667** | **0.685** | **0.497** | **0.608** |
| 90% Error, Beam=5 | **0.546** | **0.582** | **0.676** | **0.624** | **0.565** | **0.680** | **0.676** | **0.696** | **0.504** | **0.617** |

---

**Case Study: Slef-Reflection and Error-Correction During Inference Time**

**Raw prediction:**      R`<reflect>`KYFHNELM+15.995NYVQEC+57.021QFDSETSL$
**Post-processed output:** KYFHNELM+15.995NYVQEC+57.021QFDSETSL
**Ground-truth label:**   KYFHNELM+15.995NYVQEC+57.021QFDSETSL

---

**Case Study: Self-Reflection but no-op During Inference time**

**Raw prediction:**      KDFFTYME`<reflect>`E$
**Post-processed output:** KDFFTYME
**Ground-truth label:**   KDFFTYME

---

**Outperforming Bio-Inspired Models Without Domain Modules.**   As shown in Table **??**, our reflection-pretrained model achieves state of the art in peptide sequencing task, **surpassing all bio-inspired baselines**, despite using **no task-specific architectural modules**. Built on a standard Transformer, it leverages the expressive power of reflection-driven Chain-of-Thought reasoning, demonstrating that procedural reasoning can outperform hard-coded biological priors.

**Reflection Behavior Case Study.** As shown in Table of Case studies, the model exhibits **strong reflection** capabilities during inference. It successfully detects and corrects its own mistakes using `<reflect>`, and equally important, it chooses to retain correct outputs when no revision is needed. This balance between self-correction and confident continuation highlights the model's ability to exercise self-doubt as well as self-affirmation.

**Impact of Error Injection on Generalization.** Figure 5 shows the effect of different reflection error ratios on validation loss during pretraining. Without error injection (blue curve), the model exhibits clear signs of overfitting: validation loss initially decreases but begins to rise after 100 steps. In contrast, injecting reflection errors–especially at 90% and 99% rates–leads to more stable and monotonic decreases in validation loss. This improvement stems from the fact that our **batch-dynamic error** alters sequences throughout the training, preventing the model from memo-

Table 3: Reflection usage during inference (Among incorrect sequences). **Use**: % sequence uses reflection token among incorrect sequences if reflection were banned; **Corr.**: % reflections that correct the errors among all usage in incorrect predictions; **Same**: % reflections that retained the original error among all incorrect predictions. **The rest are changing one error to another error.**

| Model | Use | Corr. | Same |
|---|---|---|---|
| Baseline | 0.0% | – | – |
| Reflect FT | 0.0% | 0.0% | 0.0% |
| PT 60% Error | 10.9% | 16.0% | 69.9% |
| PT 90% Error | 16.8% | 14.1% | 65.2% |

rizing fixed input-output mappings. As a result, the model is forced to generalize across perturbed inputs, reducing overfitting.

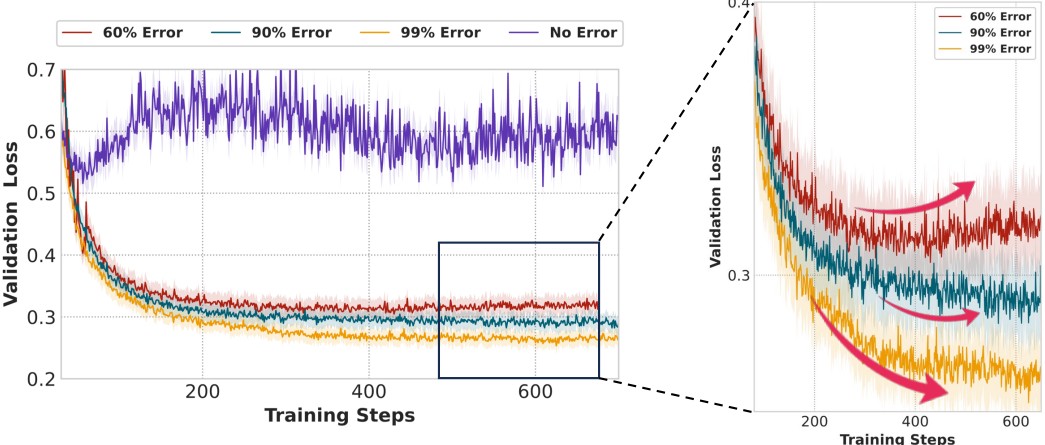

Figure 5: Error Injection Rate on Validation Loss. Loss Plotting and Logging by Neptune.ai.

**Human-in-the-Loop Reflection.** We manually picked 10 sequences where the model failed to detect errors. By manually inserting `<reflect>` tokens at the error positions and then generating from there, we found that 2 out of 10 sequences were successfully corrected during inference. While this approach is not yet scalable–especially in real-world settings lacking ground-truth labels–it highlights a promising opportunity for expert biologists to intervene and guide reflective reasoning.

Additional results, including evaluation on extended benchmark datasets, the effect of beam size, and further case studies with analysis, are provided in Appendix.

---

**Case Study: Manual Reflection Improves Prediction**

| | |
|---|---|
| **Raw prediction:** | RLANLYWL$ |
| **Manual intervention:** | RL`<reflect>` |
| **Output after intervention:** | RL`<reflect>`MNFYGFL |
| **Ground-truth label:** | RMNFYGFL |

---

**Human-Guided Reflection for Correction.** Case study in Table 4.2 demonstrates the potential of human-in-the-loop intervention using reflection tokens. As shown above, the model initially produces an incorrect prediction ("RLANLYWL"), but upon manually inserting a `<reflect>` token after the first residue, the model self-corrects and generates the correct target sequence ("RMNFYGFL").

Notably, the correction cascades beyond a single token—the model adjusts not just the second residue but the entire downstream sequence. This highlights the model's capacity for global reasoning triggered by a local reflective signal. Although we do not yet have automated methods for locating such errors in unlabeled real-world data, this opens the door for expert-guided inference: domain experts can flag uncertain regions and inject `<reflect>` tokens to improve model output. This enables a promising direction for collaborative, controllable biological sequence generation.

## 5 CONCLUSION

We introduce a reflection-based pretraining approach that equips biological sequence models with intermediate reasoning abilities, inspired by Chain-of-Thought prompting. This improves expressiveness, encourages self-correction, boosts accuracy, and reduces overfitting–particularly for *de novo* peptide sequencing. The framework also enables human-in-the-loop use, bridging biological and natural language models and highlighting the potential of reasoning-driven modeling in biology.

## 6 REPRODUCIBILITY STATEMENT

To ensure the reproducibility of our research, all experiments are conducted on the publicly available MassIVE-KB dataset (Wang et al., 2018), with evaluations performed on the standard 9-species benchmarks. Our proposed reflection pretraining framework, including the RPRE and RPLE error injection strategies, dynamic data updating, and gradient blocking, is described in detail in Section 3 with Code available in the provided link. The model is a standard Transformer encoder-decoder, and all architectural details and training hyperparameters—such as learning rates, optimizer, and hardware configuration—are specified in Section 4.1 and the Appendix. To further facilitate replication, we released our source code and model weights in the abstract section with a anonymous link, with sample reflection output also provided there. Together, these resources provide a comprehensive basis for verifying our results and extending our work.

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

## APPENDIX

## A RELATED WORK

**Chain-of-Thought Reasoning in Natural Language Models.** Traditional natural language processing (NLP) systems predominantly followed an *answer-only* generation paradigm, directly mapping inputs to final outputs. This changed with the introduction of Chain-of-Thought (CoT) prompting (Wei et al., 2022), which enables models to emit intermediate reasoning steps-non-answer tokens-prior to producing a final answer. CoT significantly improves performance on tasks requiring compositional reasoning (Nye et al., 2021; Kojima et al., 2022), by allowing models to retain transient memory, perform iterative updates, and correct earlier errors during generation. The success of CoT has led to a surge in structured reasoning approaches, including tree-structured (Yao et al., 2024; Long, 2023), graph-based (Besta et al., 2024; Sel et al., 2023), and decompositional methods (Zhou et al., 2022; Drozdov et al., 2022; Khot et al., 2022). Further extensions (Zelikman et al., 2022; Madaan et al., 2024; Suzgun & Kalai, 2024) demonstrate that the effectiveness of these methods hinges on the high expressiveness of human language-capable of articulating logic, uncertainty, abstraction, and reflection of which enrich the non-answer space.

**Limitations of Current Biological Sequence Generation.** Deep learning models have advanced sequence modeling in biology across DNA (Alipanahi et al., 2015; Zhang et al., 2021), RNA (Yang et al., 2022; Deng et al., 2022), and proteins (Rives et al., 2021; Jumper et al., 2021; Elnaggar et al., 2021; Lin et al., 2023). Yet, despite these advances, generation in biological domains-particularly in *de novo* peptide sequencing (Yilmaz et al., 2022; Eloff et al., 2023; Xia et al., 2024)-remains constrained to answer-only prediction. These models operate over token spaces (e.g., 20 amino acids) with inherently limited semantic richness, producing final sequence tokens without externalized reasoning traces. We argue that this limitation is not merely architectural but stems from the low expressive capacity of biological sequence languages. Unlike natural language, biological alphabets are poorly suited for encoding intermediate states, error attribution, or hypothetical exploration capabilities central to CoT-style reasoning.

**Expressiveness, Computational Power, and Generative Capacity.** While Transformers with CoT prompting can theoretically simulate Turing-complete machines (Pérez et al., 2021; Merrill & Sabharwal, 2023; Strobl et al., 2024), their practical reasoning capacity is constrained by input length, memory bandwidth (Arora & Barak, 2009; Garrison, 2024), and crucially, the expressiveness of the output token space. In biological models, the inability to produce diverse non-answer tokens impedes the model's ability to represent internal state transitions or conduct hypothesis-driven exploration. This constitutes a fundamental bottleneck, distinct from issues of model size or architecture. In contrast, natural language models such as GPT (Achiam et al., 2023) benefit from an output space rich in semantics, enabling a flexible reasoning substrate. Biological models like ESM (Rives et al., 2021) or ProLLaMA (Lv et al., 2025) lack this capacity, as their generation is grounded in symbolic alphabets that are not optimized for reasoning abstraction.

**Embedding Reflection and Self-Correction into Biological Models.** Recent advances in NLP promote self-reflection and critique during inference (Madaan et al., 2024), but these methods are predominantly post hoc and not integrated into model training. Our work is the first to incorporate reflection and error correction directly into the pretraining phase of biological sequence models.

Specifically, we propose a reflection-based pretraining strategy for *de novo* peptide sequencing, where models are exposed to corrupted prediction traces and explicitly trained to identify and amend errors. This goes beyond inference-time prompting, embedding reasoning patterns into model weights. Complementary efforts in scaling Transformer context (Beltagy et al., 2020; Zhang et al., 2023; Jiang et al., 2023) address sequence length limitations but do not address the deeper issue of representational constraints. By introducing a structured set of non-answer tokens and designing tasks that invoke reasoning within biological domains, our framework redefines the generative capacity of biological models to support more intelligent, interpretable, and compositional generation.

# B  MODEL EXPRESSIVE POWER: FROM MLPs TO RECURRENT ARCHITECTURES AND TRANSFORMERS

The **expressive power** of a neural network architecture dictates its fundamental capacity to represent or approximate complex functions, thereby determining its suitability for a diverse range of computational tasks. This capacity is intrinsically linked to the architectural design, especially its mechanisms for information propagation and transformation across processing stages. Architectures with constrained expressive power may falter on tasks demanding deep sequential reasoning or intricate dependency modeling, whereas more expressive models can capture a richer set of complex patterns and computational processes.

## B.1  DEFINING EFFECTIVE COMPUTATIONAL DEPTH

We propose to formalize a critical dimension of expressive power via the concept of *Effective Computational Depth*, denoted $D_{\text{eff}}(\mathcal{M})$. This metric quantifies the maximal length $L$ of a sequence of non-trivial transformations $(\mathcal{T}_i)_{i=1}^{L}$ that a model $\mathcal{M}$ (with parameters $\theta$) can compose to map an input $x$ to an output $y^*(x)$, potentially averaged over a distribution of tasks $\mathcal{P}_{\text{task}}$. Each $\mathcal{T}_i$ represents a distinct computational step, possibly conditioned on intermediate results generated by preceding transformations $\mathcal{T}_{j<i}$. Formally:

$$D_{\text{eff}}(\mathcal{M}) := \sup \left\{ L \in \mathbb{N} \; \middle| \; \exists \mathbf{T}^{(L)} \subseteq \mathcal{M} \text{ s.t. } \left( \forall P \in \mathcal{P}_{\text{task}}^{(L)}, \; \mathcal{R}_{\min}(P, \mathbf{T}^{(L)}) \le \epsilon \right) \right\} \quad (1)$$

Here, $\mathcal{P}_{\text{task}}^{(L)}$ signifies tasks inherently requiring $L$ sequential computational steps. $\mathcal{R}_{\min}$ represents the minimum achievable risk (or loss) for task $P$ using the sequence of $L$ transformations $\mathbf{T}^{(L)}$ afforded by model $\mathcal{M}$, and $\epsilon$ is a small tolerance indicating successful task resolution. A higher $D_{\text{eff}}(\mathcal{M})$ signifies a greater capacity for deep sequential processing, a cornerstone of tackling complex computations. The depth complexity, as discussed in prior work, measures the number of sequential steps after considering all parallel processing a model performs. $D_{\text{eff}}(\mathcal{M})$ aims to capture this notion of maximal sequential transformative capacity.

## B.2  MULTI-LAYER PERCEPTRONS (MLPs)

Multi-Layer Perceptrons (MLPs) are structured with a fixed number of layers, say $m$. Computation proceeds sequentially through these layers: $h^{(i)} = \sigma(W^{(i)} h^{(i-1)})$. Each layer's operation constitutes a transformation $\mathcal{T}_i$. Since $m$ is a constant, independent of the input sequence length $n$, the effective computational depth $D_{\text{eff}}(\text{MLP})$ is $O(m)$. This effectively becomes $O(1)$ with respect to $n$, as $m$ does not scale with input size. This inherently fixed and often shallow depth restricts MLPs from adeptly solving tasks that necessitate iterative refinement or the processing of sequences with variable and potentially long-range dependencies.

## B.3  RECURRENT NEURAL NETWORKS (RNNs)

Recurrent Neural Networks (RNNs) incorporate recurrent connections, enabling the output from a previous time step $h_{t-1}$ to be an input to the current step: $h_t = g_\theta(h_{t-1}, x_t)$. This architectural feature allows information to persist and be transformed across a number of time steps that can scale with the input sequence length $n$. In this context, each transformation $\mathcal{T}_i$ corresponds to the recurrent computation at a given time step. Consequently, the effective computational depth $D_{\text{eff}}(\text{RNN})$ can reach $O(n)$. This substantial increase in sequential depth capacity empowers RNNs to effectively

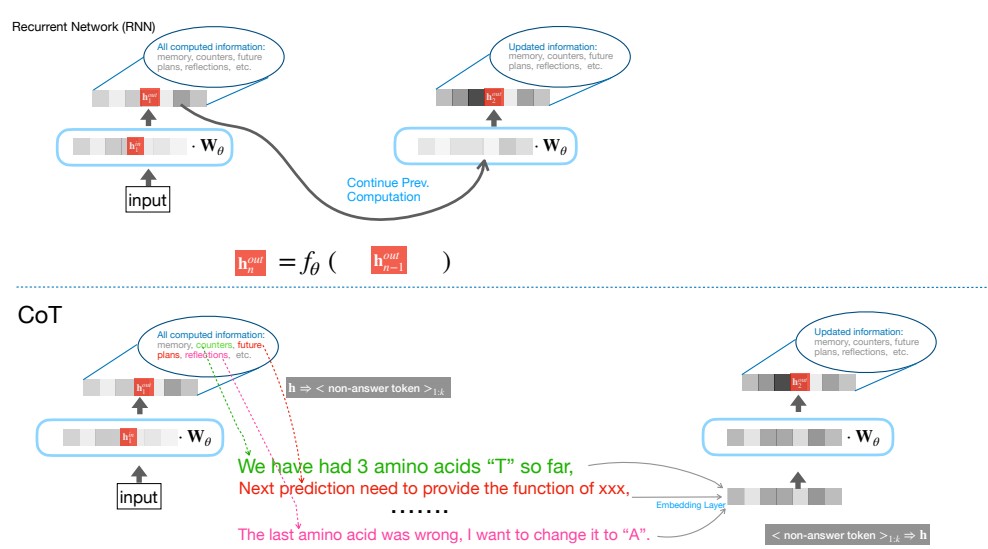

Figure 6: Illustration of how CoT approximate recurrent computation in LMs.

model sequential data and tasks characterized by temporal dependencies, such as string manipulations or sequential decision-making, far surpassing the capabilities of standard MLPs. RNNs are considered "recurrence-complete" as they possess the capability to simulate any one-term recurrent function $h_t = g'(h_{t-1})$, given adequate network capacity and appropriate non-linear activation functions, a property underpinned by the Universal Approximation Theorem.

### B.4 TRANSFORMERS

Standard Transformer architectures, despite their considerable success driven by the attention mechanism, operate with a fixed number of layers, $m$, akin to MLPs. While attention allows for parallel processing of tokens within each layer and sophisticated information integration from the entire input sequence $x_{1:t}$, the number of sequential transformation steps (i.e., layers) remains constant. The final layer output $h_t^{(m)}$ is a function of the input $x_{1:t}$ but not directly of a temporally prior hidden state $h_{t-1}^{(m)}$ in the same way as in an RNN. The transformations $\mathcal{T}_i$ in a Transformer correspond to the operations within each of its $m$ layers. Therefore, the $D_{\text{eff}}(\text{Transformer})$ is $O(m)$, which is $O(1)$ with respect to the input sequence length $n$. This characteristic limits their inherent ability to solve tasks that require a computational depth greater than $m$ without architectural modifications or prompting strategies that simulate deeper recurrence.

## C CHAIN OF THOUGHT: TOWARDS UNBOUNDED EXPRESSIVE POWER IN THEORY

The advent of Chain of Thought (CoT) prompting has been empirically shown to significantly enhance the reasoning capabilities of Large Language Models (LLMs). Beyond a mere prompting strategy, CoT can be interpreted from a computational theory perspective as a mechanism that fundamentally reshapes the Transformer's operational dynamics at runtime. In theory, CoT endows autoregressive LLMs with the potential for unbounded expressive power by dynamically altering their computational depth, time complexity, and effective memory.

### C.1 AUGMENTING COMPUTATIONAL DEPTH VIA SIMULATED RECURRENCE

Standard Transformer architectures possess a fixed number of layers, $m$, constraining their *Effective Computational Depth*, $D_{\text{eff}}(\text{Transformer})$, to $O(m)$, or $O(1)$ with respect to input sequence length

$n$. This architectural limitation inherently restricts their capacity for tasks demanding extensive sequential reasoning steps exceeding $m$.

Chain of Thought circumvents this limitation by simulating a recurrent computational process. The core mechanism involves:

1. Encoding the model's internal computational state (hidden state $h$) at the end of a reasoning step into a sequence of natural language tokens $(o_1, o_2, ..., o_k)$ that form the "thought". This can be represented as $h \rightarrow o_{1:k}$.

2. Appending these tokens to the ongoing input sequence, effectively creating an extended context $x_{n+k} = (x_1, ..., x_n, o_1, ..., o_k)$.

3. The model then processes this extended sequence, and the CoT string $o_{1:k}$ is decoded back into a computational state $h'$, effectively $o_{1:k} \rightarrow h'$.

This iterative process, $h^{(t)} \rightarrow o_{1:k}^{(t)} \rightarrow h^{(t+1)}$, emulates the recurrent connection $h_t = g(h_{t-1}, x_t)$ found in RNNs, where the model can resume and refine its computation based on the explicitly articulated intermediate state. This transformation allows the internal hidden state $h$, containing comprehensive computational information, to be first externalized into a sequence of "non-answer tokens" (the CoT string). These tokens, which can articulate ongoing reasoning, memory states, or even self-corrections, are then embedded and re-processed by the model to inform subsequent computations, thereby achieving the $o_{1:k} \rightarrow h'$ step. Consequently, the effective computational depth is no longer rigidly tied to the number of physical layers but rather to the length of the generated thought sequence, $T(n)$. The $D_{\text{eff}}(\text{LLM+CoT})$ can thus become $O(T(n))$, theoretically allowing for arbitrarily deep sequential computation, limited primarily by factors like context window length or coherence of the generated thought, rather than by fixed architectural depth. This approximated recurrence is what notably improves the model's computational capacity.

## C.2 IMPACT ON COMPUTATIONAL TIME

The simulation of recurrence through CoT inherently impacts computational time. Standard autoregressive generation processes one token at a time. By extending the input sequence with intermediate thought tokens $o_{1:k}$, the CoT process necessitates additional computational steps for both generating these thought tokens and subsequently processing them as part of the augmented input. If an input of length $n$ requires $T(n)$ CoT steps (where each step might involve generating multiple tokens), the overall time complexity for processing is increased. The time complexity is enhanced to approximately $O(n + T(n)_{\text{tokens}})$, where $T(n)_{\text{tokens}}$ represents the total number of tokens generated as part of the chain of thought. While this increases the computational cost, it is this very extension that facilitates the enhanced reasoning depth.

## C.3 EXPANDING EFFECTIVE MEMORY THROUGH ARTICULATED STATES

Transformers, through their attention mechanism, can access all previous tokens in their context window. However, the critical information from intermediate computational steps might be diffusely encoded within hidden states and not directly accessible or interpretable for subsequent, distinct reasoning phases that exceed the fixed-layer depth.

CoT provides a mechanism to externalize these intermediate computational states. By articulating the hidden state $h$ (or critical aspects of it, such as memory, counters, or reflections) into natural language strings $o_{1:k}$, these strings act as an explicit, persistent memory trace within the context window. The model can then attend to these textual representations of its prior states in subsequent steps. This allows for a form of "read-write" memory capability where the model "writes" its state as text and "reads" it back to continue computation. This capability is crucial for tasks that require maintaining and manipulating information over extended reasoning chains, effectively emulating more powerful memory structures. For instance, LLMs with CoT can achieve tape-like memory access via attention on the generated text, enabling them to tackle tasks at higher levels of the Chomsky hierarchy, such as context-sensitive tasks, which are typically beyond the reach of standard Transformers without CoT. The expressiveness of natural language is posited to be sufficiently universal to encode diverse types of information, including reasoning states, memory, and intermediate results, which is a critical assumption for this mechanism's success.

In essence, CoT theoretically empowers Transformers by allowing them to dynamically construct deeper computational graphs and manage more explicit memory traces at runtime, thereby transcending some of their inherent architectural limitations.

## D  WHY AND WHAT'S *De Novo* PEPTIDE SEQUENCING?

Peptide sequencing from tandem mass spectrometry (MS/MS) data is a cornerstone of proteomics (Aebersold & Mann, 2003a), driving advancements from fundamental biological discovery to drug development (Aebersold & Mann, 2003b; Ng et al., 2023). The goal is to determine the amino acid sequence of peptides directly from their mass spectra. Traditional database search algorithms (Eng et al., 1994; Perkins et al., 1999; Cox & Mann, 2008; Zhang et al., 2012), though widely used, cannot identify novel peptides absent from reference databases. This limitation impedes progress in emerging applications such as *de novo* antibody characterization (Beslic et al., 2022), neoantigen discovery (Karunratanakul et al., 2019), and metaproteomics (Hettich et al., 2013). In contrast, *de novo* peptide sequencing infers sequences without relying on any database, making it indispensable for comprehensive and unbiased proteomic analysis.

The advent of deep learning (LeCun et al., 2015) has transformed *de novo* sequencing, surpassing traditional algorithmic approaches (Dančík et al., 1999; Ma et al., 2003; Frank & Pevzner, 2005). Early models such as DeepNovo (Tran et al., 2017) employed CNNs and LSTMs, while more recent efforts have shifted to Transformer-based architectures (Vaswani et al., 2017), which offer improved scalability and performance. These models generally fall into two categories: autoregressive (AT) and non-autoregressive (NAT). AT models—such as Casanovo (Yilmaz et al., 2022; 2023) and its derivatives (e.g., AdaNovo (Xia et al., 2024), HelixNovo (Yang et al., 2024), InstaNovo (Eloff et al., 2023), ContraNovo (Jin et al., 2024))—predict one amino acid at a time, often achieving high precision through architectural refinements. NAT models like PrimeNovo (Zhang et al., 2024b), on the other hand, generate sequences in parallel, enabling bidirectional context and faster inference (Gu et al., 2017; Xiao et al., 2023). Despite this progress, both paradigms largely lack the capacity for explicit intermediate reasoning, which may limit generalization in noisy or uncertain scenarios.

In this work, we propose *Reflection-Pretraining*, a novel pretraining strategy that enables models to generate intermediate reasoning steps, bringing CoT-style capabilities to domains beyond natural language. Rather than predicting the final output directly, *Reflection-Pretraining* encourages the model to emit structured intermediate tokens that guide and justify the final prediction. By applying this method to the *de novo* sequencing setting, we demonstrate that structured reasoning improves generalization, robustness, and interpretability across peptide prediction tasks.

We identify *de novo* **peptide sequencing** as an ideal task for exploring structured neural reasoning for five key reasons:

- It is a foundational problem in proteomics, as peptide sequencing remains the primary method for identifying naturally occurring amino acid sequences.
- The task is inherently reasoning-intensive, requiring interpretation of complex spectral patterns based on biochemical rules, well aligned with the Chain-of-Thought (CoT) paradigm.
- Large-scale, high-quality training datasets of spectra paired with peptide sequences are readily available.
- Evaluation is exact and unambiguous, with each spectrum matched to a ground-truth sequence, unlike more subjective tasks such as language modeling or structure prediction.
- Each predicted token can represent a distinct reasoning step in the sequence inference process, enabling fine-grained supervision and reflection.

These properties make the task uniquely suited to evaluate models that go beyond final-answer generation and instead perform stepwise reasoning under uncertainty.

In the *de novo* peptide sequencing task, the model is given a spectrum instance $\mathbf{H} = \{\mathbf{I}, c, m\}$, produced by a mass spectrometer. The spectrum consists of:

- $\mathbf{I} = \{(\text{m/z}_1, i_1), (\text{m/z}_2, i_2), \ldots, (\text{m/z}_k, i_k)\}$, a set of $k$ observed mass-to-charge and intensity pairs, filtered by a signal threshold.

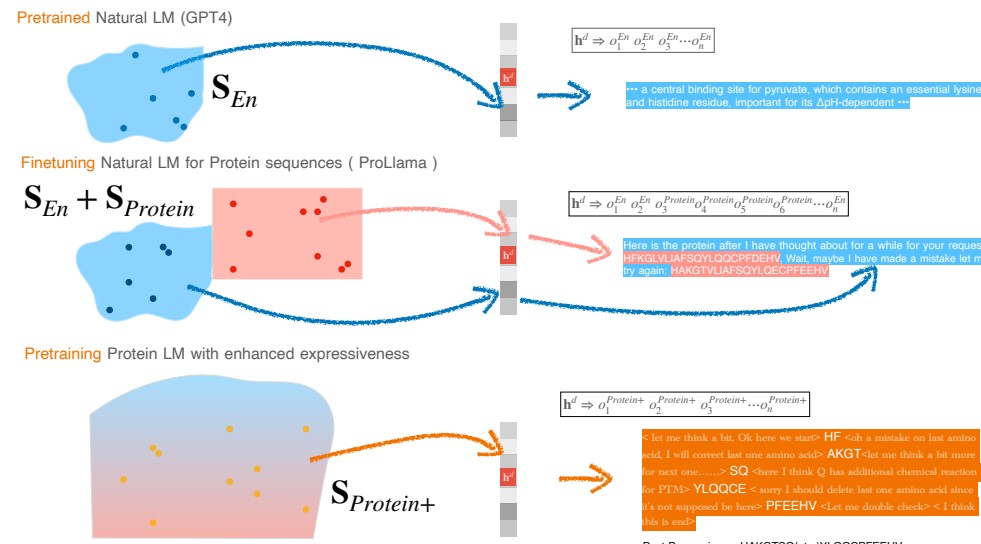

Figure 7: Illustration of the disjoint expressive spaces between natural language (NL) and protein sequences under fine-tuning. CoT reasoning capabilities learned in $L_{\text{NL}}$ do not transfer into $L_{\text{protein}}$, resulting in no shared reasoning ability.

- $c \in \mathbb{Z}^+$, the charge state of the precursor ion, and
- $m \in \mathbb{R}^+$, the total measured mass of the peptide.

The objective is to **predict the underlying amino acid (protein) sequence $\mathbf{A} = \{a_1, a_2, \ldots, a_n\}$**, where each token $a_i \in \mathcal{V}_{\text{protein}} = \{20 \text{ amino acids tokens}\}$ belongs to the vocabulary of the protein language $L_{\text{protein}} = (\mathcal{G}_{\text{protein}}, \mathcal{V}_{\text{protein}})$. The mapping $\mathbf{H} \mapsto \mathbf{A}$ requires the model to reason over the structure and intensity patterns in $\mathbf{I}$ while conforming to biochemical constraints imposed by $c$ and $m$.

*Note that, although we describe our method in the context of the protein sequence space, it applies to other biological sequence prediction tasks-including RNA, DNA, and synthetic polymers.*

## E WHY FINETUNING FAILS TO TRANSFER REASONING TO PROTEIN SEQUENCES

Let $L_{\text{NL}}$ denote the natural language space (e.g., English), and $L_{\text{protein}}$ denote the protein language space, comprising amino acid tokens with limited compositional structure. Large language models (LLMs) such as LLaMA or GPT-4 are pretrained to reason over $L_{\text{NL}}$ and can perform advanced tasks through CoT prompting—generating intermediate reasoning steps, storing temporary variables, performing error correction, and maintaining self-reflection, all in natural language.

In approaches like **ProLlama**, a pretrained natural language model is fine-tuned on protein sequences, introducing a new vocabulary $\mathcal{V}_{\text{protein}}$ and dataset distribution $\mathbb{S}_{\text{protein}} \subset \mathcal{X}_{\text{protein}}^*$. While the transformer architecture remains shared, the representations fine-tuned on proteins diverge significantly from those learned in natural language. As illustrated in Figure 7, the model's learned expressiveness splits into two largely non-overlapping subspaces:

$$\mathbb{S}_{\text{NL}} \cap \mathbb{S}_{\text{protein}} \approx \emptyset \tag{2}$$

This results in a model that performs well on surface-level protein generation but cannot leverage its natural language reasoning skills—because those abilities remain confined to the $L_{\text{NL}}$ space and are inaccessible in $L_{\text{protein}}$.

To understand this intuitively, consider a bilingual LLM pretrained on English. If we fine-tune it on Chinese texts without joint training or alignment, the model learns to generate Chinese tokens fluently, but cannot reason about Chinese using English thinking patterns. For example, tokens like "let me double check" or "I made a mistake" have no meaningful grounding in the Chinese token space unless the model was trained to use them cross-lingually. The model ends up with two islands of knowledge—English and Chinese—that cannot communicate or share reasoning capacity.

The same phenomenon occurs in protein modeling. ProLlama retains its CoT reasoning ability in English but cannot express or apply that reasoning in $L_{\text{protein}}$. As a result, it cannot generate intermediate reasoning tokens like `<reflect>` or revise outputs mid-generation—capabilities that are essential for self-correction, debugging, or guided sequence exploration.

In contrast, our reflection-pretraining approach explicitly introduces auxiliary reasoning tokens $\mathcal{V}_{\text{reflect}}$ and trains the model to use them within the protein generation process. This expands the language from $L_{\text{protein}}$ to an augmented form:

$$L_{\text{protein+}} = (\mathcal{G}_{\text{protein}}, \mathcal{V}_{\text{protein}} \cup \mathcal{V}_{\text{reflect}}) \tag{3}$$

and correspondingly enlarges the expressive space from $\mathbb{S}_{\text{protein}}$ to $\mathbb{S}_{\text{protein+}}$, allowing the model to generate and interpret internal reasoning sequences natively in the protein domain:

$$|\mathbb{S}_{\text{protein+}}| \gg |\mathbb{S}_{\text{protein}}| \tag{4}$$

## F    REFLECTION TOKEN FOR DIFFERENT ERROR CORRECTION

In this section, we provide intuitive examples of how the proposed `<reflect>` token functions as a backtracking mechanism to enable error correction in biological sequence generation. The reflection token allows the model to explicitly mark a mistake and revise it, rather than committing to an incorrect sequence. We illustrate three common error types: **deletion**, **swapping**, and **modification**.

### F.1    DELETION CORRECTION

**Ground truth:**

```
M--K--L--T--P
```

**Wrong prediction:**

```
M--K--T--P
```

(here residue `L` was mistakenly skipped)

**With reflection:**

```
M--K--<reflect>--insert(L)--T--P
```

The reflection token explicitly signals the error and recovers the missing `L`.

### F.2    SWAPPING CORRECTION

**Ground truth:**

```
A--G--S--V
```

**Wrong prediction:**

```
A--S--G--V
```

(`G` and `S` are swapped)

**With reflection:**

```
A--S--<reflect>--swap(G,S)--G--V
```

The reflection token backtracks, flags the swap, and restores the correct order.

### F.3 MODIFICATION CORRECTION

**Ground truth:**



`M--R--Q--H`



**Wrong prediction:**



`M--R--E--H`



(`Q` was incorrectly predicted as `E`)

**With reflection:**



`M--R--<reflect>--replace(E,Q)--Q--H`



The reflection token enables the model to mark the erroneous substitution and replace it with the correct amino acid.

### F.4 SUMMARY

These examples show how the `<reflect>` token acts like a biological "backspace" or "edit marker," allowing the model to recover from local mistakes. Unlike standard sequence generation—which commits to an error once produced—reflection-based generation provides a structured way to annotate, revise, and correct intermediate predictions. This ability is crucial for biological sequences, where even single-residue errors can have drastic structural or functional consequences.

## G EXPERIMENTS

**Model Training.** We adopt the same architecture and optimization configuration as in prior work of training Transformer for De novo Peptide Sequencing. All inputs—including MS/MS peaks, precursor features, and amino acid tokens—are first embedded into a shared 400-dimensional latent space.

The model is built on a 9-layer Transformer, with each layer comprising 8 attention heads and a 1024-dimensional feedforward network. This architecture forms the backbone for both the encoder and decoder components.

Training is conducted with a batch size of 1600 and an initial learning rate of 4e-4. The learning rate linearly warms up during the first epoch before decaying via a cosine schedule. Optimization is performed using AdamW (Kingma & Ba, 2014), and all models are trained for 30 epochs on eight A100 GPUs.

These hyperparameters—embedding dimension, transformer depth, attention head count, and learning rate policy—are kept fixed across all experiments unless explicitly stated. Additional implementation details can be found in our released codebase.

**Base Bio-inspired Model Designs.** We evaluate our approach against a range of bio-inspired peptide sequencing models, each of which incorporates domain-specific knowledge or architectural modifications tailored to biological data:

- **Peaks** (Ma et al., 2003) employs a protein database search strategy that combines tandem mass spectrometry with homology-based alignment. This approach excels in navigating ambiguity in peptide identification using prior biological information.
- **DeepNovo** (Tran et al., 2017) combines convolutional and recurrent neural networks with peptide-specific preprocessing and adopts a species-wise cross-validation scheme.
- **PointNovo** (Qiao et al., 2021) builds on point-cloud processing ideas to handle varied-resolution spectra without increasing model complexity, showcasing robustness through biologically-informed design.
- **InstaNovo** (Eloff et al., 2023) introduces a diffusion model that improves performance through iterative refinement of predicted sequences.
- **HelixNovo** (Yang et al., 2024) extends CasaNovo by modifying the MS/MS input to explicitly encode complementary b/y ion relationships—information that is biologically grounded in peptide fragmentation rules.

In contrast to these models, our reflection-pretrained Transformer introduces no biological heuristics or domain-specific preprocessing. Instead, it augments a standard model's reasoning capacity via token-level reflection and self-correction. This purely cognitive enhancement allows the model to outperform bio-inspired baselines without incorporating any explicit biochemical priors.

**Analysis of Beam Size Influence.** We examined the effect of beam search width on the performance of the Reflection Pretrain model. As shown in Table 2, increasing the beam size consistently improves model performance across both Amino Acid (AA) Precision and Peptide Recall metrics. Specifically, AA Precision increased from $0.788$ at beam size 1, to $0.797$ at beam size 3, and further to $0.806$ at beam size 5. This reflects an absolute improvement of $0.018$, corresponding to a relative increase of approximately $2.28\%$. A similar trend was observed for Peptide Recall, which rose from $0.600$ (beam size 1) to $0.608$ (beam size 3), and reached $0.617$ at beam size 5—an absolute gain of $0.017$ and a relative improvement of approximately $2.83\%$. These results suggest that larger beam sizes enable the model to explore a broader hypothesis space during decoding, thereby reducing the risk of prematurely discarding promising sequence candidates. Nevertheless, the performance benefits must be balanced against the increased computational cost in both time and memory—an inherent trade-off in beam search.

## H    THE USE OF LARGE LANGUAGE MODELS (LLMS)

Large Language Models (LLMs) served as assistive tools for improving the clarity and grammar of our academic prose. Specifically, we leveraged GPT-5 for drafting and refining sections such as the introduction and method. The authors retain full responsibility for all scientific content, including the conception of the research questions, methodological contributions, and the validation of experimental results.