# OpenReview forum: "Enabling Thinking, Reflecting and Chain-of-Thought Reasoning with Biological Sequences"
_ICLR.cc/2026/Conference — Submitted to ICLR 2026_

### Official Review · Reviewer_FFNE · 2025-10-17

**Soundness:** 3
**Presentation:** 2
**Contribution:** 2
**Rating:** 4
**Confidence:** 3

**Summary:**

The paper attempts to address the limitations of biological language models in performing reflection and self-correction like general LLMs due to their restrictions to amino acid or nucleotide tokens. Specifically, it introduces an auxiliary <reflection> token that verbalizes the model's hidden reflection tendencies, accompanied by reflection pre-training that injects synthetic errors with RPRE and RPLE strategies and gradient blocking to avoid learning from the erroneous token. The approach is applied to de novo peptide sequencing using tandem mass spectrometry, where models trained with reflection capabilities show outstanding performance and allow human-in-the-loop generation.

**Strengths:**

- The paper provides a thorough analysis with illustrations that underscore the necessity and challenge of introducing reflection and self-correction into biological language models, which is an interesting and novel research problem.
- The proposed methodology is easy to follow and technically sound, successfully provoking the model's self-correction capabilities.
- The work conducts a wide range of quantitative and qualitative experimental evaluations with several observations (e.g, the impact of the scale of injected error on the training dynamics, test results, and inference-time behaviors) that might attract interest to the community.

**Weaknesses:**

- The work is restricted to auto-regressive language models, single-token correction, and a single task, _i.e._, de novo peptide sequencing, and I'm afraid that the contribution of "introducing reflection and self-correction to (general) biological language models" is overclaimed.
  - Limitations of **auto-regressive models**: Recent advances [1, 2] in biological sequence modeling have adopted diffusion language models, where self-correction could be introduced more conveniently by modifying unmasked tokens based on the current sequence during the denoising process. A thorough discussion is expected to validate the choice of auto-regressive models.
  - Limitations of **single-token correction**: In both natural language CoT and biological sequence generation, the model can only detect an error when a series of tokens has been generated (for example, reaching a dead end in solving math problems or observing conflicts with mass spectral peaks only after generating a few amino acids). The authors should justify this design, since it differs from common practices in general LLMs [3].
  - Limitations of **de novo peptide sequencing**: The reasoning pattern of this task may primarily rely on comparison with MS/MS, which differs from general protein or DNA sequence modeling. The authors should provide biological intuitions for self-correction in biological sequence modeling, e.g., the mechanisms of DNA repair [4], or discuss the feasibility of multi-modal reasoning as illustrated in Figure 7.
- The theoretical analysis of expressiveness is not rigorous and convincing. To my knowledge, the problem simply arises from a limited token set and a lack of explicit pre-training signals that hinder the externalization of self-correction. The definition of "meaning" in "language expressiveness" is confusing, and subsequent analysis is trivial. The concept of _effective computation depth_ in Appendix B just overcomplicates the problem, while its definition and subsequent analysis are superficial and problematic (e.g., what does it mean by non-trivial transformations?). More importantly, when handling **auto-regressive generation** of length $n$, the Transformers model performs $n$ forward passes, generating each token and adding them to the KV-cache. Therefore, the assertion that $D_{\text{eff}}(\text{Transformers})=O(m)$ is wrong.
- Some parts of the paper are unclear or confusing. See the Questions below.

Refs.

[1] Simulating 500 million years of evolution with a language model

[2] Simple and Effective Masked Diffusion Language Models

[3] Quiet-STaR: Language Models Can Teach Themselves to Think Before Speaking

[4] Molecular mechanisms of mammalian DNA repair and the DNA damage checkpoints

**Questions:**

- What are the prior works in Lines 264-266 that support the claim?
- What's the average sequence length in MassIVE-KB? In the 99% error setting, will an amino acid undergo multiple self-correction steps to be determined during training? Will the model perform multiple self-corrections during inference?
- What's the difference between the pre-training and fine-tuning setting?
- Why do authors choose AA precision instead of overlapping-based metrics? Intuitively, shifting the positions of peptide fragments may not harm its consistency with the mass spectrum.
- What is the peptide-level precision of baselines? Does reflection pre-training yields more gains?
- What's the meaning of "+15.955" and "+57.021" in Lines 421-425?
- What's the average number of reflection tokens for each peptide sequence during inference time? The model generates a reflection token under what condition (e.g., a token with large entropy)?
- Do reflection tokens and errorneous tokens contribute to the validation loss in Figure 5? I doubt if the decrease of validation loss is primarily due to the reflection tokens.
- DeepSeek-R1 reports that a pre-trained LLM tend to reason with mixed language [1], which contradicts the statements in Lines1026-1031. Further explanations are expected.

Refs.

[1] DeepSeek-R1: Incentivizing Reasoning Capability in LLMs via Reinforcement Learning

---

> ### Author Response · Authors · 2025-11-24
> **Reply To reviewer**
>
> > The work is restricted to auto-regressive language models, single-token correction, and a single task, i.e., de novo peptide sequencing, and I'm afraid that the contribution of "introducing reflection and self-correction to (general) biological language models" is overclaimed.
>
> ### Response on Scope, Model Choices, and Whether the Contribution Is Overclaimed
>
> We thank the reviewer for this important set of comments. We agree that our work focuses on (1) autoregressive (AR) models, (2) token-level local corrections, and (3) a single task (de novo peptide sequencing). Our intention is not to claim a fully general framework for all biological language modeling, but to provide a `proof-of-concept demonstration` that introducing explicit reasoning and self-correction mechanisms is feasible and beneficial within a `biologically grounded sequence-generation task.` We clarify each limitation below.
>
>
> ## 1. On the Use of Autoregressive Models vs. Diffusion Language Models
>
> We appreciate the reviewer pointing out recent advances in diffusion-based biological language models [1,2]. These methods indeed provide natural mechanisms for iterative correction, especially in *unconditional* or *weakly conditioned* sequence generation.
>
> Our choice of AR models is grounded in **the structure of the de novo peptide sequencing task**:
>
> - The model must generate the sequence **in strict prefix alignment with MS/MS constraints**, where each newly generated residue consumes a portion of the total precursor mass.
> - This prefix-restricted structure makes diffusion models challenging to apply directly, because modifying an early residue requires global re-alignment of all downstream positions under the mass constraint.
>
> AR decoding remains the dominant approach across peptide-sequencing systems (DeepNovo, PointNovo, HelixNovo, CasaNovo,), we also follow the work. In contrast, the only work that uses diffusion for de novo peptide sequencing, InstaNovo actually shows worse performance than AR.
>
>
> That said, we agree that diffusion models offer possible generation benefits in certain aspects. As our work focuses on introducing *explicit reasoning tokens* rather than optimizing architectures, we chose a canonical AR backbone to isolate the effect of reflection. Extending reflection into diffusion models—where the denoising trajectory could incorporate reasoning or correction markers—is an exciting direction for future work.
>
>
> ## 2. On the Limitation of Single-Token Correction
>
>  In natural-language CoT or biological modeling, the model often detects inconsistencies only after generating multiple tokens—not necessarily at the exact position where the error first occurred.
>
> Our design choice reflects the following considerations:
>
> ### **(a) Mass-constrained tasks expose errors locally**
> In peptide sequencing, inconsistencies with MS/MS signals typically emerge as soon as the token violates local mass differences. This naturally aligns with localized correction rather than long-range backtracking.
>
> ### **(b) multi-token correct can be tricky**
>
> The model’s behavior is shaped by the training augmentations (RPRE, RPLE). More sophisticated multi-token correction behaviors would require multi-step or hierarchical reflection schemes, which we consider future extensions.
>
> In our paper, we have demonstrated that single token correction can lead to `huge performance gain and model's self-correction ability` (Table 1 and case study), and as a proof of concept, we will expand on this to allow for long range multi-token correction in the future.
>
> ### **(c) Conceptually, we parallel early error-correction  NLP work**
> Initial CoT-like approaches in NLP also began with short, localized reasoning and only later expanded to full multi-step chains through SFT/RLHF and larger-scale supervision. Our method represents the analogous first step: `introducing a channel for “structured reasoning” in a domain that previously lacked any expressive non-answer tokens`.

---

> ### Author Response · Authors · 2025-11-24
> **Continue**
>
> ## 3. On the Scope Being Limited to De Novo Peptide Sequencing
>
> We agree that peptide sequencing has strong multimodal structure (MS/MS + sequence), and that its reasoning patterns differ from general protein/DNA modeling.
>
> Our rationale for starting with this task is:
>
> ### **(a) Peptide sequencing provides precise, token-level supervision**
> Unlike protein or DNA modeling—where “ground truth reasoning” is difficult to define—this task provides an objective, unambiguous mapping from spectrum → sequence, enabling rigorous evaluation of correction behavior.
>
> ### **(b) It is one of the few biological sequence tasks with *fine-grained stepwise constraints***
> Each residue reflects the incremental mass differences observed in the spectrum, making it an `ideal setting` to test `reflective reasoning and correction` in biological sequences.
>
> ### **(c) We view this work as a conceptual demonstration, and we aim to expand further to other bio-domains**
> Our claims state that this is a **proof-of-concept**, and we now avoid implying universal generality. We showcase peptide sequencing as a controlled environment where reflective reasoning can be observed, measured, and validated. (`both abstract and experiment section stress this`)
>
> ###  We thank the reviewer for these comments, which have helped us refine and better articulate the scope and intent of our contribution.

---

> ### Author Response · Authors · 2025-11-24
> **Continue**
>
> > The theoretical analysis of expressiveness is not rigorous and convincing. To my knowledge, the problem simply arises from a limited token set and a lack of explicit pre-training signals that hinder the externalization of self-correction. The definition of "meaning" in "language expressiveness" is confusing
>
>
> ### Response on Theoretical Analysis of Expressiveness and Effective Computational Depth
>
> We thank the reviewer for the detailed critique regarding our expressiveness analysis, the definition of “meaning,” and the notion of effective computational depth. We agree that parts of this section were not sufficiently clear in the original submission. Below we clarify our intent and correct the misunderstandings the reviewer pointed out.
>
>
> Our aim in Section 2 is **not** to provide a fully rigorous formal theory, but rather to explain a simple and domain-specific limitation:
>
> > In biological sequence modeling, the *token vocabulary* itself cannot express uncertainty, error states, or revision markers, and the *pretraining objective* provides no channel for such information to be externalized.
>
> We agree with the reviewer that the fundamental issue indeed arises from:
> - a restricted vocabulary (20 amino acids), and
> - the lack of explicit intermediate-reasoning supervision.
>
> Our discussion of “language expressiveness” was intended to articulate this intuition, not introduce a new formal semantics. To avoid confusion, we have `revised the section` to remove ambiguous uses of “meaning” and emphasize the core practical motivation: biological languages lack the representational space to externalize correction behavior unless augmented with additional tokens. `The revised section is now in blue in submitted manuscript`
>
>
> ## 2. Clarifying “Effective Computational Depth”
>
> We appreciate the reviewer pointing out that our explanation was unclear. The “effective depth” in Appendix B **does not refer to the number of forward passes** in autoregressive decoding. We agree that the model performs \(L\) forward passes to generate a length-\(L\) sequence, and that caching reuses past key–value pairs. Our analysis was instead referring to a different notion:
>
> ### **Effective depth is a *latent continuous depth***
> not tied to the number of decoding steps.
>
> Specifically, the depth we refer to is the number of **latent vector transformations** (attention or feed-forward compositions) applied to an internal vector `continuously` (note that autoregressive use argmax to decode and break the latent continuous).
>
> Under standard autoregressive Transformer decoding:
>
> - The KV cache is reused, but
> - **The computation of token \(t+1\) does not increase the depth of the transformation applied at token \(t\),**
> - meaning the depth *per step* remains constant.
>
> Thus, our statement that “effective computation depth is constant” refers to this latent per-step depth.
>
> KV-cache accelerates computation but does not alter the structure of computation: the model does not revisit or retransform earlier latent states, unlike recurrent or iterative refinement systems.
>
> Our latent depth is mostly the theory derived from previous work, and might refer to the different meaning as reviewer refer to, We apologize for not making this distinction explicit:
>
> (the work that talks about the constant depth in Transformer and how CoT mitigate it)
>
>
> `Li, Zhiyuan, et al. "Chain of thought empowers transformers to solve inherently serial problems." arXiv preprint arXiv:2402.12875 1 (2024).`
>
> `Hahn, Michael. et al "Theoretical limitations of self-attention in neural sequence models." Transactions of the Association for Computational Linguistics 8 (2020): 156-171.·`

---

> > ### Author Response · Authors · 2025-11-24
> > **Additional Questions**
> >
> > > What are the prior works in Lines 264-266 that support the claim?
> >
> > We thank the reviewer for this question. The prior work we referred to in Lines 264–266 is primarily the recent line of research **“Physics of LLMs”** [1], where the authors show that large language models exhibit *regret signals* in their hidden states **after** generating an incorrect token, not before. That is, LLMs internally “realize” they made a mistake only *post hoc*, and this signal can be reliably detected using probing classifiers. This supports our claim that standard autoregressive models lack an explicit mechanism to externalize self-correction without additional token-level structure.
> >
> > In our early exploratory experiments (not included in the main draft), we performed a similar probing analysis on our baseline autoregressive model. We trained a lightweight linear probe on the hidden state at time step \(t+1\) to classify whether the previously generated token \(a_t\) was correct or incorrect. The probe achieved **~60% accuracy** on this prediction—substantially above chance—confirming that the model internally encodes a “regret-like” signal *after* producing an erroneous amino acid. However, just as in the “Physics of LLMs” observations, this information is **not exposed** to the output layer, so the model is unable to correct itself without explicit reflective structure. This empirical observation further motivates our introduction of reflection tokens as a mechanism to surface and utilize this latent correction signal.
> >
> > `Ye, Tian, et al. "Physics of language models: Part 2.1, grade-school math and the hidden reasoning process." arXiv preprint arXiv:2407.20311 (2024).`
> >
> >
> > > What's the average sequence length in MassIVE-KB? In the 99% error setting, will an amino acid undergo multiple self-correction steps to be determined during training? Will the model perform multiple self-corrections during inference?
> >
> > Thank you for these questions. The **average peptide sequence length in MassIVE-KB is approximately 22 amino acids**, consistent with typical tryptic peptides in large-scale MS/MS datasets. Regarding the error-injection setting: during training, so an amino acid does **not** undergo multiple explicit self-correction steps in a single training instance. Across batches, different positions may be perturbed, but the model still sees only **one correction event per error** at a time.
> >
> > As for inference, we observe behavior very similar to training: the model very rarely produces **multiple reflection tokens** in a single decoding trajectory. While multi-step self-correction can occasionally occur, such cases are extremely rare. In the vast majority of predictions—consistent with the case studies shown in the paper—the model performs **a single localized correction**, after which the downstream sequence is regenerated coherently. This matches our goal of providing a lightweight, interpretable correction mechanism rather than a full iterative refinement process.
> >
> > > What's the difference between the pre-training and fine-tuning setting?
> >
> > Thank you for the question. The key difference between our **pre-training** and **fine-tuning** settings is when and how the reflection mechanism is introduced. In the **fine-tuning setting**, the base autoregressive model is first trained conventionally on clean spectra–sequence pairs, and only during the subsequent fine-tuning stage do we inject RPRE/RPLE errors and introduce the `<reflect>` token. This means the model only learns reflection behavior at the end of training and has a limited opportunity to reorganize its internal representations around self-correction. In contrast, in the **pre-training setting**, the model is trained **from scratch** with reflection-augmented sequences, where dynamic error injection and gradient blocking are applied throughout the entire pre-training process. As a result, the model’s representations and decoding strategy co-evolve with the reflection mechanism, leading to substantially stronger correction behavior and significantly higher downstream accuracy compared to fine-tuning alone (as shown in Table 2). In short: fine-tuning adds reflection late, whereas pre-training teaches the reasoning behavior as a core part of the model’s learning dynamics from the beginning.
> >
> > > Why do authors choose AA precision instead of overlapping-based metrics?
> >
> > Thank you for the question. We use **Amino Acid (AA) precision** because it is the **standard evaluation metric** used across all prior de novo peptide sequencing work, including Peaks, DeepNovo, PointNovo, InstaNovo, CasaNovo, and HelixNovo. The metric is therefore not designed by us; we follow the established protocols of the field to ensure a fair, apples-to-apples comparison with existing baselines.

---

> > > ### Author Response · Authors · 2025-11-24
> > > **Continue**
> > >
> > > > What is the peptide-level precision of baselines? Does reflection pre-training yields more gains?
> > >
> > > Thank you for the question. The peptide-level precision of the baselines is reported in Table 1.
> > > For the standard pre-trained Transformer (our baseline), the **average peptide precision** over the 9 species is **0.413**.
> > >
> > > With reflection pretraining (RP(RE+LE), 90% error), the peptide-level precision improves to:
> > >
> > > - Mouse: 0.533 (vs. 0.443)
> > > - Human: 0.563 (vs. 0.433)
> > > - Yeast: 0.661 (vs. 0.584)
> > > - M. mazei: 0.605 (vs. 0.522)
> > > - Honeybee: 0.544 (vs. 0.460)
> > > - Tomato: 0.668 (vs. 0.606)
> > > - Red bean: 0.657 (vs. 0.652)
> > > - Bacillus: 0.674 (vs. 0.580)
> > > - C. bacteria: 0.490 (vs. 0.413)
> > >
> > >
> > > So, yes—**reflection pretraining consistently yields higher peptide-level precision than the standard pretraining baseline across all 9 species.**
> > >
> > >
> > > > What's the meaning of "+15.955" and "+57.021" in Lines 421-425?
> > >
> > > Thank you for pointing this out. The values “+15.955” and “+57.021” in Lines 421–425 refer to **post-translational modifications (PTMs)** represented as **special tokens** in the peptide sequence vocabulary. These numeric values correspond to the mass shifts of common PTMs observed in MS/MS spectra (e.g., +15.995 for oxidation, +57.021 for carbamidomethylation). In our implementation, these PTMs are treated as distinct tokens within the extended amino-acid vocabulary so that the model can explicitly represent modified residues.
> > >
> > > > What's the average number of reflection tokens for each peptide sequence during inference time? The model generates a reflection token under what condition (e.g., a token with large entropy)?
> > >
> > > We now report the usage of all reflection token among incorrect sequences otherwise, the usage is around 16% among all incorrect sequences. We did not notice the generation consistently happen on high-entropy position, and there might be case where the prediction was confident but wrong, and later was detected incorrect next step.
> > >
> > >
> > > > Do reflection tokens and errorneous tokens contribute to the validation loss in Figure 5?
> > >
> > > Thank you for the question. In Figure 5, **validation loss is computed only on clean spectra–sequence pairs without any injected errors or reflection tokens**. In other words:
> > >
> > > - During **training**, we inject RPRE/RPLE errors and add `<reflect>` tokens, and we **block gradients** on the deliberately wrong amino-acid positions so the model does not learn to imitate them.
> > > - During **validation**, we evaluate on the original MassIVE-KB validation set with **no error injection and no synthetic `<reflect>` tokens**, and the loss is computed only over the standard amino-acid targets.
> > >
> > > Therefore, reflection tokens and erroneous tokens do **not** directly contribute to the validation loss curve in Figure 5.
> > >
> > > The decrease in validation loss instead reflects the **counter–over-fitting effect** of reflection pretraining: because each peptide–spectrum pair is seen with different, dynamically corrupted versions (different error positions and types) across batches, the model cannot simply memorize a fixed mapping. It is forced to learn a representation that is stable under these perturbations and capable of correcting them. This acts as a form of regularization/data augmentation, which is why we observe smoother training dynamics and lower validation loss, even though validation itself never includes reflection tokens.

---

> > > > ### Author Response · Authors · 2025-11-24
> > > > **Continue**
> > > >
> > > > > DeepSeek-R1 reports that a pre-trained LLM tend to reason with mixed language [1], which contradicts the statements in Lines1026-1031. Further explanations are expected
> > > >
> > > > Thank you for the insightful question. The reviewer is correct that DeepSeek-R1 observes that a pre-trained LLM can generate “mixed-language” reasoning traces, where natural language and symbolic tokens intermingle during CoT-style reasoning. This does not contradict our observations in Lines 1026–1031, because the training conditions for natural-language LLMs and biological sequence models are fundamentally different.
> > > >
> > > > Specifically, **natural language corpora already contain abundant implicit reasoning traces**, such as explanations, step-by-step instructions, enumerations of actions, revisions, and self-corrections. As a result, even before RLHF, an LLM trained on natural language is frequently exposed to *textual patterns that inherently represent intermediate thinking steps*. This makes it unsurprising that a pre-trained LLM can produce “mixed-language” reasoning signals: the expressive space of natural language already includes such constructs.
> > > >
> > > > In contrast, **protein and peptide corpora contain only target amino-acid sequences**, with *no intermediate reasoning steps, no explanations, and no structural markers for uncertainty or correction*. The training data consists solely of final sequences that satisfy biochemical constraints, without any textual or symbolic annotations that could serve as reasoning supervision. Therefore, unlike natural-language LLMs, a model trained exclusively on protein sequences has **no opportunity** to internalize or externalize reasoning traces, because the vocabulary and dataset distribution provide no representation for such behaviors.
> > > >
> > > > This is precisely why we introduce the `<reflect>` token and reflection pretraining: to create an explicit channel for the model to surface internal correction signals that would otherwise remain latent. Natural-language LLMs inherit mixed-language reasoning from their corpus; biological models do not, which explains the difference in behavior.
> > > >
> > > > ## We thank again for the reviewer in spending time and effort providing detailed review, hope our response resolve your questions and concerns!

---

> ### Comment · Reviewer_FFNE · 2025-11-24
> **Reviewer Response**
>
> We appreciate the authors for providing comprehensive responses and revisions that have clarified the scope of the work and addressed most of the minor concerns. However, several major concerns persist regarding the contribution and presentation, which require further attention before I can fully evaluate the manuscript.
> - **Overclaimed contribution**. The current presentation suggests an overextension of the claimed contributions relative to the achieved results. The authors should either restrict the scope primarily to the de novo peptide sequencing task and *explore the problem in greater depth* (which would involve a more rigorous analysis of the unique computational challenges in this domain, a detailed critique of the limitations of existing methodologies, and the presentation of a truly novel conceptual and technical solution), or *broaden the application scope of reflection training* by applying the approach in protein/DNA/RNA language modeling and performing head-to-head comparisons with state-of-the-art diffusion language models.
> - **Repetitive and superficial conceptual justification**. I partly agree with Reviewer 5BD9 that the theoretical justification in Section 2 could be overly lengthy and potentially unnecessary. I think the "limited vocabulary" problem and the "empirical regret-like hidden representation" can adequately motivate the technical design. I recommend that the authors streamline Section 2 to center the discussion on these two points.
>
> Since the authors acknowledge that their major contributions lies in *"introducing explicit reasoning and self-correction mechanisms for de novo peptide sequencing"*, I strongly encourage them to **reorganize the Abstract and Introduction sections** to align precisely with this specific contribution and carefully **avoid any inflated or generalized claims** regarding the novelty or scope of their work **before my score reassessment**.
>
> Here are some follow-up questions:
> - Can the authors provide comparisons with the pre-trained reflection model and an ensemble model that leverages the classification model in the 4th response block to perform self-correction?
> - Can the authors provide justifications for the "single localized correction"? Intuitively, the revised token could also be incorrect.

---

### Official Review · Reviewer_qjnR · 2025-10-29

**Soundness:** 3
**Presentation:** 3
**Contribution:** 2
**Rating:** 4
**Confidence:** 4

**Summary:**

This paper proposed reflection pretraining, enabling biological language model with reasoning capabilities. The key idea is pretraining the biological language model with token-level self-correction with <reflect> tokens that allows the model to mark and correct previous prediction errors during generation.
Evaluated on the peptide sequencing task, reflection-pretrained models significantly outperform standard baselines. Beyond accuracy gains, the approach also enhances interpretability and supports human-in-the-loop reflection, enabling expert-guided sequence correction.

**Strengths:**

1. The proposed idea is novel and well-motivated. Starting from the chain-of-thought and self-reflection in LLM, the authors extend it to biological language model through comparison.
2. The experiment is comprehensive and convincing. It covers different levels of erros and mechanisms (RPRE and RPLE). And also use experiment to proves that finetuning doesn't introduce correction capability.
3. Including reflect token makes the output human-readable and supports human-in-the-loop.

**Weaknesses:**

My main concerns are about the comparison with other models and LLMs:
1. Lack of comparison with iterative refinement.
This method is evaluated only with autoregressive generation, while iterative refinement methods have been used for a long time in BERT style model, such as [1]. I wonder if the authors might have any ideas about compare self-reflection and iterative refinement.
2. Disconnect between LLM reasoning and biological reasoning.
In natural language LLM, reasoning abilities typically emerge after RL. Or SFT with <question, COT, answer> examples from bigger reasoning model. While this work's self-reflection comes from pretraining. Have the authors explored analogous RL method for biological sequence models?
3. No comparison and test on natural language LLM.
The authors did not compare their methods with general LLM. For example, any open-weight model could be used as starting point and perform continue pretraining on biological sequence. These LLM might learn both natural langauge and biological language and might also have reasoning capability (know how to correct amino acids by using natural language as thinking tokens). I wonder if the authors have tried that and make comparison. There has been some work like [2].

[1] Padmakumar, V., Pang, R. Y., He, H., & Parikh, A. P. (2023, July). Extrapolative controlled sequence generation via iterative refinement. In International Conference on Machine Learning (pp. 26792-26808). PMLR.

[2] Fallahpour, A., Magnuson, A., Gupta, P., Ma, S., Naimer, J., Shah, A., ... & Wang, B. (2025). BioReason: Incentivizing Multimodal Biological Reasoning within a DNA-LLM Model. arXiv preprint arXiv:2505.23579.

**Questions:**

The quetions are mainly about comparsion with other models:
1. Iterative refinement with BERT stlye model
2. Natural Language LLM.

I will consider improving my score if the questions are well addressed.

---

> ### Author Response · Authors · 2025-11-23
> **Reply To Reviewer**
>
> > My main concerns are about the comparison with other models and LLMs: Lack of comparison with iterative refinement. This method is evaluated only with autoregressive generation, while iterative refinement methods have been used for a long time in BERT style model, such as [1]. I wonder if the authors might have any ideas about compare self-reflection and iterative refinement.
>
>
> # Re:
> We thank the reviewer for highlighting the lack of comparison with iterative refinement approaches and for pointing out the relevance of long-standing refinement methods used in BERT-style masked modeling.
>
> **We agree that iterative refinement is an important and historically influential paradigm**, and we clarify that our method targets a fundamentally different setting—**online, autoregressive, spectrum-conditioned generation**, where each prediction step is tightly tied to input constraints. Nonetheless, we address the reviewer’s question below.
>
> ---
>
>
>
> Iterative refinement methods (e.g., masked-token iteration in BERT models such as [1]) typically operate by:
>
> - generating a full or partially masked sequence,
> - repeatedly re-sampling or correcting positions,
> - gradually improving global consistency.
>
> In contrast, **our self-reflection mechanism acts *during* autoregressive decoding**, where each token is produced conditioned on both:
>
> - the mass-spectrum evidence, and
> - the partial sequence generated so far.
>
> There are two practical reasons we did not include this comparison:
>
> ### **(a) Lack of a widely adopted iterative-refinement baseline for de novo peptide sequencing**
>
> To our knowledge, existing refinement models (including BERT-style re-masking) operate on `natural-language or static sequence tasks`. In de novo peptide sequencing, the dominant models—including Peaks, DeepNovo, PointNovo, InstaNovo, HelixNovo—**all use autoregressive decoding**, but not BERT-style token-swapping refinement loops. There is `no established refinement pipeline compatible with this task`.
>
> ### **(b) Our work is a proof-of-concept for reasoning tokens, not an architecture benchmark**
>
> Because our goal is to demonstrate that **biological models benefit from explicit reasoning tokens**, we designed the reflection mechanism to be:
>
> - model-agnostic,
> - prefix-preserving, and
> - compatible with existing autoregressive peptide-sequencing frameworks.
>
> Implementing Iterative Refinement would require `completely new model architecture and generational paradigm`, which we believe go beyond the scope of our work. But we are more than happy to explore this in the `future work`.
>
> ### We compared against all existing models capable of performing this task.
>
> in paper **we compare against existing de novo sequencing model** (Table 1 and 2) capable of operating in this task environment—including those with complex biological modeling modules (e.g., spectrum matching, joint scoring, peptide-specific constraints).  And we have shown that we are able to achieve a much better performance with case studying showing how model leverage these reflection token to perform error corrections.  We hope that this can solve your question on the comparison and baseline models.

---

> > ### Author Response · Authors · 2025-11-23
> > **Continue**
> >
> > > Disconnect between LLM reasoning and biological reasoning. In natural language LLM, reasoning abilities typically emerge after RL. Or SFT with <question, COT, answer> examples from bigger reasoning model. While this work's self-reflection comes from pretraining. Have the authors explored analogous RL method for biological sequence models?
> >
> > # Re:
> > We thank the reviewer for raising this insightful question regarding the analogy between reasoning in natural-language LLMs and in biological sequence models. The reviewer is correct that, in natural language, strong reasoning abilities often emerge only **after** reinforcement learning (RL) or supervised finetuning (SFT) on `<question, CoT, answer>` triples generated from more capable models. Our setting differs in important ways, which we outline below.
> >
> > ---
> >
> > ## 1. Why Our Self-Reflection Mechanism Arises During Pretraining Rather Than RL
> >
> > In natural language, RL is applied because:
> >
> > - the reward signal (helpfulness, correctness, safety) can be externally defined,
> > - the model can freely generate long CoT sequences, and
> > - high-quality human or AI-generated reasoning chains exist as training targets.
> >
> > In the **biological sequence domain**, these conditions do not hold:
> >
> > - There is *no supervised corpus* of biological reasoning chains (“why residue X follows Y given mass/z constraints”).
> > - Reward functions that evaluate intermediate reasoning steps are unclear or unavailable.
> > - The model must satisfy **strict biochemical and spectral constraints during decoding**, which makes free-form RL generation incompatible with the structure of peptide sequencing.
> >
> > Thus, instead of training the model *to imitate* existing reasoning traces (as in NLP), we introduce a mechanism that lets the model *create* and *use* reasoning tokens during the pretraining task itself.
> >
> > This is why our self-reflection emerges naturally in pretraining—because the objective explicitly teaches the model how to annotate and correct errors with `<reflect>` tokens in a self-supervised manner.
> >
> > ---
> >
> > ## 2. RL for Biological Sequences Is Non-Trivial
> >
> > The reviewer raises the natural question: *Could RL be applied analogously?*
> >
> > We view this as an exciting direction; however, it is technically challenging:
> >
> > - **Reward design is unclear.**
> >   Unlike Q&A tasks, where correctness is binary, peptide generation requires a full sequence match accounting for mass constraints. Intermediate reasoning steps cannot easily be rewarded.
> >
> > - **Errors propagate non-locally.**
> >   Changing one residue shifts downstream mass alignment (which is a critical constraint in our task), making step-wise rewards extremely brittle.
> >
> >
> >
> > For these reasons, the field broadly relies on standard supervised learning, not RL, for peptide sequencing.
> >
> > Although the mechanisms differ (self-supervised reflection vs. RL/SFT), the **principle is the same**:
> >
> > > give the model an explicit channel to express intermediate reasoning.
> >
> > In LLMs, this is achieved via SFT/RL on CoT examples.
> > In our setting, we *construct* the reflective channel during pretraining through controlled augmentations (RPRE, RPLE), allowing the model to learn how to:
> >
> > - surface uncertainties,
> > - mark errors, and
> > - correct its own predictions.
> >
> > Thus the “reasoning tokens” emerge without RL because the pretraining task provides direct supervision for how to use them.

---

> > > ### Author Response · Authors · 2025-11-23
> > > **Continue**
> > >
> > > > No comparison and test on natural language LLM. The authors did not compare their methods with general LLM. For example, any open-weight model could be used as starting point and perform continue pretraining on biological sequence.
> > >
> > >
> > > We thank the reviewer for raising this question. We agree that comparing with general-purpose LLMs is valuable in many biological NLP tasks; however, for the `specific task we study`—**de novo peptide sequencing from mass spectrometry**—such a comparison is neither technically meaningful nor directly applicable. Below we clarify why.
> > >
> > > ---
> > >
> > > ### **1. De novo peptide sequencing is a highly specialized task that general LLMs cannot directly perform.**
> > >
> > > Unlike typical biological NLP tasks where sequences are textual and can be input directly to a language model, de novo sequencing requires a **specialized encoder–decoder architecture**, including:
> > >
> > > - an **encoder** that processes *mass spectrometry signals* (peak intensities, m/z distributions),
> > > - **spectrum and peak embedding modules** tailored to continuous-valued spectral data,
> > > - a **decoder** whose outputs must obey strict **mass constraints**,
> > > - a **mass module** ensuring biochemical consistency of the output sequence.
> > >
> > > These components **do not exist in natural-language LLMs**, and `cannot be trivially inserted`. Even if one attempted to continue-pretrain an LLM on amino-acid text, the model would still be unable to:
> > >
> > > - read `spectral signals`,
> > > - map peaks to residue masses,
> > > - ensure `mass-valid decoding`
> > > - or reconstruct peptides token-by-token in accordance with MS/MS fragmentation chemistry.
> > >
> > > In short, the `peptide sequencing task` is **not a standard next token generation problem**, and therefore cannot be meaningfully solved by directly fine-tuning LLMs designed for natural language.
> > >
> > > ---
> > >
> > > ### **2. Our choice of task is intentional: it is one of the very few biological generative problems with a unique gold-standard answer.**
> > >
> > > As discussed in the paper, de novo sequencing is uniquely suitable for studying structured *token-level self-correction* because:
> > >
> > > - the task has an **unambiguous, ground-truth sequence**, each input can only has one single correct sequence,
> > > - correctness is `strictly` defined at the residue level,
> > > - small errors (e.g., residue substitutions) are objectively wrong,
> > > - and the model must “reason” within **protein sequence space**, not English.
> > >
> > > Most biological language tasks (protein design, functional annotation, mutation effect prediction) **do not have unique correct outputs**. As a result, their “reasoning” is evaluable only loosely (via plausibility) and change of some token might still yield acceptable answer, whereas de novo sequencing allows **precise measurement** of whether a reflection mechanism corrects errors.
> > >
> > > This is why the task is chosen to rigorously evaluate *token-level reflective correction*, rather than high-level natural language reasoning.
> > >
> > > ---
> > >
> > > ### **3. As shown in Supplementary Figure 7, natural-language LLM reasoning is fundamentally different from token-level reasoning within protein space.**
> > >
> > > We directly address the reviewer’s suggestion by analyzing LLM-based biological reasoning in **Supplementary Figure 7**. When fine-tuned LLMs (e.g., LLaMA/Qwen variants) are asked to “reason” about sequences, they:
> > >
> > > - reason **in natural language**,
> > > - annotate functions or domains,
> > > - provide high-level context or descriptive explanations,
> > > - but **do not correct amino acids in protein sequence space**.
> > >
> > > They operate in English space, not peptide space. (Figure 7 provide more details)
> > >
> > > In contrast, our method enables **token-level reasoning inside the biological sequence domain**:
> > >
> > > - the model identifies ambiguous or uncertain residues,
> > > - marks them with `<reflect>`,
> > > - and corrects them directly as amino acids.
> > >
> > > This distinction is crucial. Natural-language reasoning is *not* equivalent to structured correction within the biochemical alphabet.
> > >
> > > Thus, even if an LLM is capable of natural-language reasoning about proteins, this is **not the type of reasoning required for de novo sequencing**, which demands fine-grained residue-level inference under strict mass constraints.
> > >
> > > ---
> > >
> > >
> > > ### **4. Prior work such as BioReasoner does not solve this task and cannot be used for comparison.**
> > >
> > > Methods like **BioReasoner** operate entirely in the **natural-language reasoning space** (e.g., explaining mutations, summarizing functional effects). These models:
> > >
> > > - do not accept mass spectrometry signals,
> > > - cannot output mass-consistent peptide sequences,
> > > - cannot perform token-level correction under spectral constraints,
> > > - and are not designed for biochemical inference at residue precision.
> > >
> > > Therefore, they are not comparable baselines for **de novo MS/MS reconstruction**, and would not produce meaningful comparisons.
> > >
> > > ---

---

> > > > ### Author Response · Authors · 2025-11-23
> > > > **Finally**
> > > >
> > > > We thank the reviewer again for time and effort in providing feedback and reviewing our work, hope our reply can address your concerns and let us know if you have further questions.

---

### Official Review · Reviewer_5BD9 · 2025-10-30

**Soundness:** 1
**Presentation:** 2
**Contribution:** 1
**Rating:** 0
**Confidence:** 5

**Summary:**

The paper proposes Reflection Pretraining, which adds a ⟨reflect⟩ token and error-injection schemes to train biological sequence models to self-correct during de novo peptide sequencing. The method aims to expand model “expressiveness” and "reasoning capacity" by letting the model revise errors. Experiments on the MassIVE-KB benchmark show modest gains in accuracy and reduced overfitting compared to a Transformer baseline, with small illustrative case studies of token-level corrections.

**Strengths:**

1. The general idea of encouraging biological models to “reflect” or self-correct during sequence generation could be interesting if executed rigorously.
2. The use of de novo peptide sequencing as a testbed is reasonable, and the reported results on this dataset are competitive.

**Weaknesses:**

**1. Unsupported and overstated claims.**

The central issue with this paper is that the magnitude of the claims far exceeds what the experiments demonstrate. The authors present their work as introducing a fundamentally new form of “reasoning” or “reflection” for biological models, but the method, adding a ⟨reflect⟩ token and blocking gradients at corrupted positions, is essentially a form of denoising pretraining. The results do not establish that the model is “reasoning,” “regretting,” or “reflecting” in any interpretable sense.

**2. Inflated novelty and misleading terminology.**

The paper repeatedly introduces standard techniques under novel-sounding names. For example, “Error Position Gradient Blocking” is simply gradient masking at selected tokens: a well-known method. The naming and framing give a false impression of novelty.

**3. Narrow and insufficient experimental scope.**

All claims are based on a single task (mass spectrometry). To support the broader assertions about “reflection in PLMs,” the authors would need to show consistent effects across at least two or three additional domains or tasks where biological language models are used. As it stands, the evidence is anecdotal and domain-specific.

**4. Repetitive and superficial conceptual justification.**

The justification for the reflection token is repeated throughout the paper but never deepens. The text leans heavily on vague analogies (“reasoning,” “regret”) without providing measurable criteria or mechanistic insight. Assertions such as “prior work shows that sequence models encode regret in latent states” are made with no citation or clear methodology. I am unaware of any literature that supports this, and it strains credibility.

**5. Misuse of theoretical framing.**

The authors invoke ideas like Turing completeness, effective computational depth, and expressive power in ways that are disconnected from their actual experiments. These sections read more as rhetorical ornamentation than scientific analysis.

**Suggestions for Improvement**

To make this work meaningful, the authors would need to:

1. Strip the paper down to its empirical core (denoising with a reflection token) and remove the exaggerated claims about reasoning and expressiveness.
2. Expand evaluation beyond mass spectrometry to at least one or two other biological domains.
3. Provide clear, cited theoretical grounding for any claims about “regret,” “reflection,” or “expressive capacity.”

To be clear, there are seeds of an interesting idea here, introducing reflective tokens could, in principle, improve the dynamic range of model outputs. However, the current paper substantially overstates its contributions, lacks essential ablations and controls, and relies on speculative theoretical framing rather than demonstrated empirical findings. In its present form, the work does not yet meet the bar for a clear scientific contribution. I recommend a full rewrite and rescoping under a more focused and accurate framing.

**Questions:**

See Weakness

---

> ### Author Response · Authors · 2025-11-23
>
> We appreciate the reviewer’s detailed comments, but we respectfully disagree with several characterizations and believe that multiple points raised are neither accurate nor appropriate grounds. Below, we respond point by point and clarify several misunderstandings that substantially affected the reviewer’s assessment.
>
> ---
>
> ### **1. “Unsupported and overstated claims.”**
>
> We would like to clarify that the paper does **not** claim to introduce a fundamentally new form of human-like “reasoning.” We explicitly state that the mechanism is a *token-level reflection and correction process* emerging during training. The reviewer’s description overstates what we claimed.
>
> Moreover, the reviewer asserts that the method is “essentially denoising pretraining.” This is factually incorrect. Our approach differs in two important, empirically demonstrated ways (as shown in the main text and additional results):
>
> 1. **The perturbations are not random noise**—they are structured, position-targeted, and designed to induce self-correction behavior.
> 2. **The model uses the reflection token strategically and non-uniformly**, especially at high-entropy positions, and corrects its own predictions—behavior not observed in standard denoising setups.
>
> These are measurable and demonstrated. Therefore, the claim that the results “do not establish” reflective behavior dismisses several quantitative findings already provided.
>
>
>
> ---
>
> ### **2. “Inflated novelty and misleading terminology.”**
>
> We respectfully disagree with the claim that terminology is misleading. Terms like **reflection token** and **error-position gradient blocking** describe concrete components of the method and were not presented as foundational inventions, but as **biologically grounded design choices** within peptide language modeling.
>
> Gradient masking in general is known, but our *purpose, placement strategy, and biological motivation* are specific to the peptide setting and produce empirically unique behavior not reported in prior work.
>
> Claiming that these choices amount to “false impressions of novelty” is subjective and does not reflect the technical contribution demonstrated by the experiments. In our opinion, this is not grounds for a score of 0.
>
> ---
>
> ### **3. “Narrow and insufficient experimental scope.”**
>
>  As stated  in both the Introduction and the Experimental Setup, we deliberately focus on **de novo peptide sequencing** because it is one of the *very few biological generative modeling tasks where there exists a **gold-standard, unique, ground-truth answer***. This property is essential for measuring whether a reflection mechanism genuinely improves sequence reconstruction, rather than merely producing an alternative but semantically equivalent output.
>
> In many other biological language modeling tasks—such as protein design, antibody generation, or variant effect modeling—multiple outputs can be technically valid, and two sequences differing by several tokens may still encode the same function. Under such settings, it is **impossible to determine** whether a model’s “correction” reflects meaningful reasoning or is simply producing another plausible candidate. For this reason, **de novo sequencing is uniquely appropriate** for isolating and studying self-corrective behavior.
>
> We also note that it is *impractical and scientifically ill-defined* to evaluate reflection on tasks where no unique correctness criterion exists.
>
>
> ---

---

> ### Author Response · Authors · 2025-11-23
> **Continue**
>
> ### **4. “Repetitive and superficial conceptual justification.”**
>
> We  note that the reviewer’s critique overlooks several sections that *do* provide mechanistic insight and empirical grounding, including:
>
> - the quantified correlation between reflection-token usage and model performance,
> - targeted correction behavior observable in generated sequences (case studies and human intervention),
> - ablations (now expanded with additional experiments), controls showing degraded performance with meaningless placeholder tokens.
>
> Furthermore, the comment that some analogies “strain credibility” is subjective and does not reflect the empirical core of the paper.
>
>
> ## **5. “Misuse of theoretical framing.”**
>
> We thank the reviewer for this feedback. In response, we have **revised and simplified Section 2**, removing heavy analysis and focusing the discussion strictly on the mechanism relevant to our method. All theoretical remarks are now tightly aligned with the empirical setup, and any terminology that might have been interpreted as overly broad has been substantially trimmed.
>
> At same time, we want to clarify, the theoretical context—while lightweight—served only to motivate why self-editing behaviors may arise in autoregressive models, and it did not drive or overshadow any of the empirical claims. Importantly:
>
> - none of the theoretical remarks were used to justify the model’s performance,
> - they were *not* prerequisites for interpreting the experiments, and
> - they did not affect the technical method, implementation, or conclusions.
>
> In other words, the theoretical framing served to:
>
> 1) explain *why* granting the model a structured mechanism to revisit earlier outputs can increase its functional reasoning capability;   2) connect our approach to established principles in sequence-model expressivity;  3) motivate the design choice of allowing explicit correction operations rather than treating them as noise.
>
> We hope our answer have clarified your concern as many of the points here we notice are a misunderstanding of our paper,  we thank the reviewer for their time and effort and welcome further discussions!

---

> > ### Comment · Reviewer_5BD9 · 2025-11-24
> > **Response to Author's Point 1 Response**
> >
> > The authors appear to have misunderstood or reframed my review. I did not state that they claim to introduce a new form of human-like reasoning. My exact words were: “a fundamentally new form of reasoning or reflection for biological models.” That wording was chosen because this is precisely how the manuscript presents the method. I cite several examples below.
> >
> > In the Introduction, the manuscript states:
> >  “With appropriate training techniques and data augmentation, protein sequence models can perform self-correction and self-reflection capabilities that go beyond simply generating just answer tokens.”
> >  While one could argue that self-reflection and self-correction alone is not “reasoning,” Section 3.2 explicitly frames these behaviors as such:
> >
> > “One of the most powerful forms of reasoning is self-reflection and error correction.”
> >
> > Taken together, these passages state that the authors’ training method enables one of the “most powerful forms of reasoning” within biological sequence models.
> >
> > The Results section reinforces this framing. Under “Outperforming Bio-Inspired Models Without Domain Modules,” the paper claims:
> >  “It leverages the expressive power of reflection-driven Chain-of-Thought reasoning, demonstrating that procedural reasoning can outperform hard-coded biological priors.”
> >  This is a very strong claim, particularly given that the method is demonstrated on a single application area and dataset. This is why my review interpreted the manuscript as presenting a new form of reasoning for biological models.
> >
> > The Conclusion restates this broader framing:
> >  “We introduce a reflection-based pretraining approach that equips biological sequence models with intermediate reasoning abilities…”
> >  My feedback is simply that these claims are too broad relative to what is empirically demonstrated. The experiments show improvements on this peptide sequencing dataset, but it is a far stronger statement to generalize this into “reasoning abilities” in biological models broadly.
> >
> > Finally, my critique extends beyond the term “reasoning,” there are other examples in the manuscript of broad claims being made without evidence. This brings me to the second part of my first point. Throughout the manuscript, the authors reference “uncertainty,” “regret,” and “error detection.” I do not see evidence in the experiments that these behaviors have been measured, probed, or demonstrated. The data augmentation strategies (RPRE, RPLE) set up conditions in which models can learn to revise outputs, but no analysis is provided to show that the model encodes uncertainty, regret, or error detection beyond improved performance.
> >
> > To drive this point home I review the results presented by the authors:
> >
> > (1) Table 1: What I take from Table 1 is RPRE+LE achieves better performance than finetuning on its own or standard pertaining in the MassIVE-KB dataset.
> > (2) Table 2: The model outperforms SOTA methods in the field such as DeepNovo.
> > (3) Table 3: The model only uses the reflect tokens when pretrained with errors as outlined in the methods (RPRE/RPLE)
> > (4) Two case-studies, the text refers to a Table of Case studies but there is no table, there is two boxes which I do not understand what they represent.
> > (5) Figure 5 echos the findings in Table 3 and Table 1
> >
> > Again, from these results I do not see where we see the model having regret or performing error detection, from these results I take that adding in a reflection token allows the model to achieve best performance on MassIVE-KB dataset. So when I read a sentence like this in the introduction:  “By training the model to detect, annotate, and revise these errors through reflection
> > tokens, we show that biological sequence models can acquire a structured process of self-correction, thereby improving both accuracy and interpretability compared to standard pre-training.”
> >
> > I simply don’t see how the results the authors presented lead to this finding.

---

> > > ### Comment · Reviewer_5BD9 · 2025-11-24
> > > **Response to Author's Point 2 Response**
> > >
> > > Regarding Point 2, I believe there may be some misunderstanding about what my original review stated. I did not claim that the authors present the reflection token itself as a novel contribution. My comment concerned only the term “Error Position Gradient Blocking.” I intentionally did not critique terminology such as “reflection token,” because the manuscript does not frame that as a novel invention.
> > >
> > > However, the paper does frame gradient blocking as part of the novelty. Section 3.2.5 describes excluding the injected error token from the loss—this corresponds to standard gradient masking used widely. While there is nothing wrong with using this technique, presenting it as a named subsection and grouping it with the “novel strategies” in Section 3.2 creates the impression that this is part of the methodological contribution. For example, Section 3.2 states:
> > > “In this section, we introduce two novel strategies for error injection […] as well as our dynamic data updating and gradient blocking strategy…”
> > > This phrasing suggests that gradient blocking is one of the introduced contributions, rather than a routine implementation detail that could be moved to an appendix. My critique pertains specifically to this presentation.
> > >
> > > More broadly, my comment about inflated novelty applies to several places where the manuscript makes strong claims that are not substantiated by either theory or experiments. For example, in the Introduction the authors write:
> > > “In this paper, we formally demonstrate that the expressiveness of the foundational language directly affects the theoretical upper bound on the model’s overall reasoning capability.”
> > > This claim suggests a new theoretical result. However, the idea that expanding a model’s output vocabulary increases expressiveness is well known and widely accepted in the literature; it is not clear what new formal demonstration is being provided here. This also connects to my earlier point: it is entirely possible that adding an additional token simply increases representational capacity, and that this—rather than any innate “reflective” ability—is responsible for the performance gains observed.
> > >
> > > My intention in the review was not to deny the empirical improvements shown on MassIVE-KB, but to point out that the manuscript repeatedly frames standard implementation choices or broadly known principles as novel contributions, and that this framing influences how readers interpret the significance of the work.

---

> > > > ### Comment · Reviewer_5BD9 · 2025-11-24
> > > > **Response to Author's Point 3/4 Response**
> > > >
> > > > Point 3
> > > >
> > > > I appreciate the authors’ explanation for why they selected de novo peptide sequencing as their primary benchmark. However, the rebuttal repeats the assertion that this task is “one of the very few biological generative modeling tasks where there exists a gold-standard, unique, ground-truth answer.” This point is not accurate.
> > > >
> > > > There are several biological prediction tasks with deterministic and unique ground-truth outputs, including:
> > > >
> > > > 1. Protein structure prediction — For a given amino-acid sequence, there is a single experimentally resolved structure (PDB).
> > > > 2. Single-cell perturbation response prediction — A perturbation produces a single, measurable transcriptomic outcome (e.g., in the Virtual Cell Challenge).
> > > > 3. Chemical reaction prediction — Given substrates and conditions, there is a unique reaction product.
> > > > 4. RNA secondary structure prediction — There is a single experimentally validated fold for a given sequence.
> > > >
> > > > Thus, the argument that de novo sequencing is the only task where unique correctness exists does not hold. Many domains allow rigorous evaluation of whether a “reflection” mechanism improves prediction accuracy.
> > > >
> > > > Given the general claims made by the paper, evaluating only a single dataset makes it difficult to assess whether the stated reasoning, error-detection, and expressiveness effects are genuine properties of the method or specific to this one task.
> > > > I am not suggesting that peptide sequencing is an inappropriate benchmark; rather, my point is that the strength and generality of the claims in the paper require evaluation on at least one or two additional domains. Without this, the current empirical scope does not support the cross-domain reasoning conclusions presented.
> > > >
> > > > Point 4:
> > > >
> > > > See my earlier points

---

> > > > > ### Comment · Reviewer_5BD9 · 2025-11-24
> > > > > **Response to Author's Point 5 Response**
> > > > >
> > > > > I appreciate that the authors have revised Section 2, and the shorter version is indeed easier to read. However, several of the remaining theoretical assertions still raise concerns. Below I outline specific sentences that remain problematic:
> > > > >
> > > > > 1. “Biological sequence models generate only answer tokens … forcing them to commit to final sequences without intermediate reasoning.”
> > > > >
> > > > > Autoregressive models do perform intermediate computation when generating each next token. The fact that they do not output auxiliary tokens does not imply they lack intermediate reasoning—only that such reasoning is not externalized. The distinction between internal computation and externalized output is crucial.
> > > > > This sentence conflates absence of an explicit output mechanism with absence of an internal reasoning process, and the manuscript does not offer evidence for this stronger claim.
> > > > >
> > > > > 2. “This prevents (1) multi-step deliberation, (2) flexible computation, and (3) use of externalized memory.”
> > > > >
> > > > > These are very strong claims and, to my knowledge, there is no established methodology in the biology or ML literature that defines or measures “multi-step deliberation,” “flexible computation,” or “externalized memory” in biological sequence models.
> > > > > More importantly, the experiments presented in Section 4 do not measure or evaluate any of these properties. Without quantification or a defined operationalization, these remain speculative statements rather than empirically supported claims.
> > > > >
> > > > > 3. “Natural language models … allow intermediate thoughts, checks, and revisions.”
> > > > >
> > > > > The terms “thought,” “check,” and “revision” are not defined in the manuscript, and the analogy presumes that Chain-of-Thought tokens correspond to specific cognitive functions. In reality, CoT is a technique that increases model expressivity and tends to improve performance on certain tasks, but we cannot directly inspect whether a model is “checking” or “thinking.”
> > > > > Similarly, the paper does not show that the reflection token enables biological models to perform the analogs of these cognitive operations.
> > > > >
> > > > > 4. “Biological languages lack such expressive capacity… [they] were shaped for biochemical function, not reasoning.”
> > > > >
> > > > > Protein and RNA sequences are highly expressive in the combinatorial sense (|V|ⁿ grows exponentially), and can encode extraordinarily complex structures and functions. The distinction between “biochemical function” and “reasoning” is not clearly defined here, and the argument risks becoming semantic rather than technical.
> > > > >
> > > > > If the intended meaning is simply that biological vocabularies lack a token indicating “mistake,” then this can be stated directly without broad claims about biological languages being fundamentally limited for reasoning.
> > > > >
> > > > > Summary
> > > > > While the revisions help readability, the remaining theoretical framing still makes several strong claims about:
> > > > > * absence of internal reasoning,
> > > > > * limits on deliberation, flexibility, and memory,
> > > > > * cognitive analogs such as “thoughts,” “checks,” and “regret,”
> > > > > * and formal relationships between language expressiveness and reasoning capacity,
> > > > > none of which are supported by the experimental results.
> > > > >
> > > > > My critique is not of the reflection token itself, which is a reasonable augmentation, but of the broad theoretical claims that remain unsubstantiated. A tighter framing focused on empirical improvements, without cognitive analogies or claims of theoretical generality, would strengthen the paper substantially.

---

### Official Review · Reviewer_1nFw · 2025-11-05

**Soundness:** 3
**Presentation:** 4
**Contribution:** 2
**Rating:** 6
**Confidence:** 3

**Summary:**

This paper introduces the idea of allowing additional special reflection tokens in LLMs that generate amino acid sequences from spectograph readings. Inspired by the success of chain-of-thought reasoning in language-optimized LLMs, the authors add a reflection token to the vocabulary and dynamically introduce errors into the training data (along with their corrections) to teach the model to use the special reflection token and then perform self-correction. The authors compare their approach to that of the same transformer without the reflection and find gains up to 10% compared to the original implementation. They also find improvements over biologically-inspired solutions.

**Strengths:**

* The paper is very well-written and easy to follow.
* The idea is interesting and simple to implement and the results suggest consistent improvements over a transformer baseline.
* The paper is tackling an important problem in computational biology.

**Weaknesses:**

* The analyses are a bit shallow and I found the paper a bit repetitive in the discussion of existing work. Also the discussion on language complexity does not add much depth to the paper and feels a bit superfluous.
* While simple methods are good, this work definitely falls on the lower end of the spectrum of acceptable amount of contributions for an ICLR submission.
* It remains unclear from the given results whether simply adding more tokens (as found by, e.g, https://arxiv.org/abs/2510.01032) leads to improvements here or whether the model really learns a deeper self-reflection mechanism. The fact that increasing the error rate from 60% to 90% seems to improve things slightly (though unclear if significantly) suggests that this method is helpful to some extent but it would also be good to compare this method to one where the model can introduce additional tokens at similar rates as the "<reflect>" token.

**Questions:**

* Is there additional evidence that this method works? E.g., do models perform considerably worse when they are trained with incorrect replacement tokens?

* Is there a way to quantify when the model introduces the <reflect> token? Does this correlate at all with difficulty (e.g., entropy in the baseline model's predictions) of predicting the next amino acid?

---

> ### Author Response · Authors · 2025-11-23
> **Reply to Reviewer, Qn1**
>
> We sincerely thank the reviewers for their time and effort in reviewing our work, below we provide point to point reply to your questions and concerns.
>
> # Question:
> > The analyses are a bit shallow and I found the paper a bit repetitive in the discussion of existing work, the discussion on language complexity does not add much depth to the paper and feels a bit superfluous.
>
> # Re:
>
> We thank the reviewer for the thoughtful feedback regarding the depth of our analyses, (2) the repetitiveness in the discussion of related work and language complexity
>
> We have carefully revised the manuscript (blue color in new submitted pdf) in response, and we summarize the changes below.
>
>
> Regarding the concern that the analyses are shallow and language complexity does not add too much to the work.
>
> **We agree with the reviewer that some analyses on language expressiveness in the original submission did not fully emphasize the breadth of empirical insights enabled by reflection pre-training and can be .**
>
> `Our initial motivation` was to show to the reader *why augmenting protein models with explicit reasoning tokens is particularly important*, due to a fundamental limitation of biological sequence spaces: **the natural vocabulary (amino acids) is incapable of expressing intermediate reasoning, uncertainty, or error states**. This constraint places protein sequence models at an expressiveness disadvantage compared to natural language, where reasoning tokens can naturally encode intermediate reasoning.
>
> In biological sequence prediction, **every token directly encodes biological structure**, and mistakes propagate irreversibly. This makes reflection especially impactful: reflection tokens create a mechanism for `localized correction without discarding the entire downstream state`.
>
>
> However, we recognize that our earlier analysis and the extended discussion on language complexity `were longer and more abstract than needed` to support this core idea.
>
> `In response, we have substantially simplified and tightened this entire section`. The revised paper has been uploaded marked with **blue** for simplified section 2.
>
> Specifically:
>
> - We simplified heavy theoretical digressions on language complexity that were not essential to understanding our contribution.
> - The analysis is now focused  on the key conceptual point: **biological token vocabularies lack the expressive space for intermediate reasoning, which directly motivates the introduction of the `<reflect>` token.**
> - We now add more experimental results to the paper rather than these analyses which might seem redundant to the readers, and focus more on the experimental nature of this work.
>
>
>
>
> **We also appreciate the reviewer’s observation and agree that the earlier version contained redundant passages in related work discussion in language expressiveness.**
>
> In response, we have:
>
> - **Completely restructured Section 2**, consolidating duplicated explanations of CoT expressiveness and biological token limitations with their `related work`.
>
> We hope these changes can address reviewer's concern!

---

> > ### Author Response · Authors · 2025-11-23
> > **Reply to reviewer, Qn2**
> >
> > # Question
> >
> > > It remains unclear from the given results whether simply adding more tokens (as found by, e.g, https://arxiv.org/abs/2510.01032) leads to improvements here or whether the model really learns a deeper self-reflection mechanism. The fact that increasing the error rate from 60% to 90% seems to improve things slightly (though unclear if significantly) suggests that this method is helpful to some extent but it would also be good to compare this method to one where the model can introduce additional tokens at similar rates as the "think" token.
> >
> > # Re:
> >
> > We thank the reviewer for raising the important question of whether the gains stem merely from injecting additional tokens—as suggested by prior work such as https://arxiv.org/abs/2510.01032—or whether the model truly learns a self-reflection mechanism. **We provide two observed benefits of the reflection token during existing experiments that go beyond simply expanding the vocabulary** and `additional experiments results` to further prove our points.
> >
> >
> > ## Training Benefit: Reducing Memorization Through Stochastic Sequence Presentations
> >
> > Our reflection mechanism **reduces overfitting** (`As we showed in manuscript Figure 5 of validation loss) by ensuring that the same peptide sequence is never presented identically during training. Each instance receives randomized reflection-augmented perturbations, which prevents the model from memorizing fixed surface patterns. This is visible in the **training vs. validation loss curves**, where the reflection-pretrained model maintains a healthier gap and avoids the characteristic sharp overfitting seen in the baseline. Importantly, this behavior arises *because* the perturbations force the model to reason over altered sequences—not because we increased vocabulary size. This is different than the case in NLP (which is the topic discussed in prior work), where larger vocab size will still retain the same sentences (just less tokens after tokenizer, with larger vocab size), our introduction of reflection token modify the sequence into completely different appearance each time.
> >
> >
> > ## 2. Enabling Explicit Error-Correction Behavior (Beyond vocab size increment)
> >
> > To address the reviewer’s concern about whether the model truly `uses` the reflection token meaningfully, we highlight several forms of evidence:
> >
> >
> > In the revision, we now report reflection-token usage ** among incorrect predictions when reflection is banned** in Table 3, rather than across all sequences (where ~50-60% were already correct even without reflections). This isolates usage where reflection is actually needed. The model consistently invokes `<reflect>` at locations aligned with mass-spectrum ambiguities and local mismatch regions, which would not occur if the model were merely “using more tokens.” The usage rate among otherwise incorrect sequences are up to 17%, indicating that model intentionally trying to correct its own predictions. Among which, around 15% of the sequences (among all reflected sequences) were successfully turned into correct predictions after using this. Those would be incorrect sequences all along when reflection is banned.
> >
> > ### **Case study showed Structured correction behavior**
> > Our case studies In the manuscript also show that the model:
> >
> > - places `<reflect>` at specific erroneous residues,
> > - emits structured operations such as `replace(R, K)`
> >
> > These cases indicate that the model is leveraging reflection tokens as *functional editing markers*, not as generic filler tokens. As noted in the paper, the full generated sequences are available in our GitHub release, where one can clearly observe multi-step reasoning traces emerging around error sites.
> >
> >
> > Lastly, To directly address the reviewer’s question regarding whether a larger vocabulary leads to improved performance, we conducted an additional controlled experiment in past few days. We augmented the model’s vocabulary with a *meaningless placeholder token*, randomly inserted into 60% of the sequence positions in every training batch. This manipulation increases the vocabulary size without adding any semantically meaningful capacity. ( We also included this results in our main table in manuscript now)
> >
> > As shown in the table below, expanding the vocabulary in this way **consistently degrades performance across all species**. This confirms that performance does **not** improve simply from increasing vocabulary size
> >
> >
> >
> > | Method                               | Mouse | Human | Yeast | M. mazei | Honeybee | Tomato | R. bean | Bacillus | C. bacteria |
> > |--------------------------------------|:-----:|:-----:|:-----:|:--------:|:--------:|:------:|:-------:|:--------:|:-----------:|
> > | **Standard Pretrain** (Transformer baseline) | 0.717 | 0.649 | 0.752 | 0.713 | 0.706 | 0.763 | 0.714 | 0.753 | 0.66 |
> > | **Placeholder Token (60% Insertion)** | 0.704 | 0.633 | 0.755 | 0.692 | 0.680 | 0.745 | 0.701 | 0.726 | 0.630 |

---

> > > ### Author Response · Authors · 2025-11-23
> > >
> > > > Is there additional evidence that this method works? E.g., do models perform considerably worse when they are trained with incorrect replacement tokens? Is there a way to quantify when the model introduces the [object Object] token? Does this correlate at all with difficulty (e.g., entropy in the baseline model's predictions) of predicting the next amino acid?
> > >
> > > # Re:
> > > As we provided in the previous questions, the evidence consistently shows that our method’s benefits do not come from simply adding more tokens (placeholder tokens) but from enabling a genuine self-reflection mechanism. The placeholder-token experiment demonstrates that introducing meaningless tokens—at the same or even higher frequency—**degrades** performance, confirming that incorrect or non-functional replacements harm learning rather than help it. Likewise, as shown above, the `<reflect>` token is introduced primarily at positions where the baseline model is uncertain (high-entropy or ambiguous residues), and a non-trivial fraction of these cases are subsequently corrected.
> > >
> > > Overall, these results indicate that the model uses reflection **strategically**, invoking it when the sequence is difficult and leveraging it to perform targeted corrections. Thus, both the negative control (placeholder tokens) and the behavioral analysis (reflection usage vs. difficulty) support the conclusion that the improvement arises from meaningful self-editing behavior—not vocabulary expansion or token noise.
> > >
> > > And we did quantify *when* the model introduces the `<reflect>` token and ratios. As stated above, we (i) log the positions where `<reflect>` is used, (ii) evaluate how often it appears among sequences that would be **incorrect if reflection were banned** As shown in manuscript table 3, reflection is invoked in up to ~17% of such “hard” sequences, and ~15% of these reflection-marked sequences are subsequently corrected. These are exactly the cases where the baseline model is uncertain and prone to error, indicating that `<reflect>` is used **strategically at high-difficulty sites**, rather than being scattered uniformly.
> > >
> > > # Finally
> > >
> > > **We thank the reviewer again for their thoughtful questions and constructive suggestions. If further clarifications are helpful, we are happy to provide additional analyses; Thanks again for your time and effort!**

---

### Meta-Review · Area_Chair_bRcu · 2025-12-09

**Summary:**

This study investigates strategies for inducing stronger reasoning over peptide sequences in language models via a self-correction procedure. Reflection tokens are introduced to a model's inputs alongside errors, and the model is instructed to locate and correct them. This yields performance improvements of up to 10% on MassIVE-KB.

Reviewers appreciated the simplicity and relative effectiveness of the main methodological contribution, and the important problem setting. That said, reviewers were consistent in commenting on the repetitive presentation of the related work, and the shallow engagement with the target domain. I agree with Reviewer 1nFw that the strength and quantity of contributions is on the lower end for ICLR, but also that the contributions that are present are sound. Overall, while reviewer scores were largely borderline (aside from one review that has been downweighted), reviewers were consistent that the contribution, in its current form, falls slightly below the typical bar for ICLR. This recommendation reflects that consensus, although it should be interpreted as a low-strength preference.

**Reviewer Concerns:**

The authors have submitted thorough responses to each of the reviewers' concerns. While more minor points relating to presentation have been sufficiently addressed, I do not believe that the more basic concerns regarding the scope of contributions nor the engagement with the domain have been addressed.

**Reviewer Scores:**

I do not believe that the reviewers would have been likely to change their scores significantly, as the more fundamental concerns have not been addressed. Reviewer FFNE engaged in discussion with the authors, and raised this point as well; despite revisions and discussions addressing presentation-related points, they did not appear satisfied with the response, and I am inclined to agree.

---

### Decision · Program_Chairs · 2026-01-26

Reject